# Evidence of $\phi_0$-Josephson junction from skewed diffraction patterns in Sn-InSb nanowires

B. Zhang[1⋆], Z. Li[1⋆], V. Aguilar[1], P. Zhang[1], M. Pendharkar[2†],
C. Dempsey[2], J. S. Lee[3‡], S. D. Harrington[4], S. Tan[5,6], J. S. Meyer[7],
M. Houzet[7], C. J. Palmstrom[2] and S. M. Frolov[1∘]

1 Department of Physics and Astronomy, University of Pittsburgh, Pittsburgh, PA, 15260, USA
2 Electrical and Computer Engineering, University of California,
Santa Barbara, CA, 93106, USA
3 California NanoSystems Institute, University of California Santa Barbara,
Santa Barbara, CA, 93106, USA
4 Materials Department, University of California Santa Barbara,
Santa Barbara, CA, 93106, USA
5 Department of Electrical and Computer Engineering,
University of Pittsburgh, Pittsburgh, PA, 15260, USA
6 Petersen Institute of NanoScience and Engineering,
University of Pittsburgh, Pittsburgh, PA, 15260, USA
7 Univ. Grenoble Alpes, CEA, Grenoble INP, IRIG, Pheliqs, 38000, Grenoble, France.

∘ frolovsm@pitt.edu

## Abstract

We study Josephson junctions based on InSb nanowires with Sn shells. We observe skewed critical current diffraction patterns: the maxima in forward and reverse current bias are at different magnetic flux, with a displacement of 20-40 mT. The skew is greatest when the external field is nearly perpendicular to the nanowire, in the substrate plane. This orientation suggests that spin-orbit interaction plays a role. We develop a phenomenological model and perform tight-binding calculations, both methods reproducing the essential features of the experiment. The effect modeled is the $\phi_0$-Josephson junction with higher-order Josephson harmonics. The system is of interest for Majorana studies: the effects are either precursor to or concomitant with topological superconductivity. Current-phase relations that lack inversion symmetry can also be used to design quantum circuits with engineered nonlinearity.

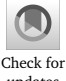

## Contents

⋆ These authors contributed equally to the development of this work.
† Current address: Department of Materials Science and Engineering, Stanford University, Stanford, CA, USA, 94305
‡ Current address: Department of Physics and Astronomy, University of Tennessee, Knoxville, TN, 37996, US

# 1 Introduction

**Context.**    Interest in superconductor-semiconductor hybrid structures is along two directions. On the one hand, they are explored as materials for quantum technologies, such as superconducting qubits [1, 2]. On the other hand, they are a platform with high potential for the discovery of topological superconductivity [3].

**Background: Josephson $\varphi_0$-junction.**    In semiconductor nanowires, a combination of induced superconductivity, spin-orbit interaction and spin splitting can famously induce Majorana modes and topological superconductivity [4, 5]. The same ingredients can induce an anomalous Josephson effect, known as $\varphi_0$-junction [6]. The primary characteristic of a Josephson junction is the current phase relation (CPR) [7]. The most common CPR is a sinusoidal function $I(\phi) = I_c \sin(\phi)$, where $I(\phi)$ is the Josephson supercurrent, $\phi$ is the phase difference between superconducting leads, and $I_c$ is the critical current. In a $\varphi_0$-junction, $I(\phi = 0) \neq 0$, which is equivalent to a phase offset $\varphi_0$ in a sinusoidal CPR [6, 8–20]. The $\varphi_0$-junction state can be accompanied by bias direction-dependent critical current [6, 14–16, 19, 20], which was dubbed the "supercurrent diode effect" [19–33].

A related effect is the $\pi$-junction effect where the additional phase shift is equal to $\pi$ [34]. Note that the $\pi$-junction is not a special case of a $\varphi_0$-junction. The $\pi$-junction can be realized under more basic conditions, for instance without any spin-orbit interaction [35].

**Challenge.**    Attempts were made to detect the phase shift $\varphi_0$ in superconducting quantum interference devices (SQUIDs) [36, 37]. However, large magnetic fields were used. These fields were in the SQUID plane in order not to induce flux in the loop. However, at high fields of hundreds of milliTesla to a few Tesla, and given the SQUID area in the tens of square microns, multiple flux quanta thread the SQUID due to fringing fields and imperfect alignment even when the utmost effort is applied to ensure strictly in-plane applied fields. Furthermore, Josephson junctions based on semiconductor nanowires are gate-tunable. This is typically thought of as an advantage due to an extra control knob. But the electric field from the gate changes the path of supercurrent in the nanowire, and with it the enclosed flux in the SQUID. A change by a fraction of a flux quantum is plausible for a 100 nm nanowire in a field of hundreds of mT. This shifts the SQUID interference pattern due to extra flux and not due to the intrinsic spin-orbit interaction and the associated $\varphi_0$-junction effect.

**Approach.**    We use supercurrent diffraction patterns as means of investigating the $\varphi_0$-junction state [19, 21]. The diffraction pattern is the evolution of the critical current in magnetic field. It can reveal exotic effects such as d-wave superconductivity in corner junctions [38], the presence of edge states in planar junctions [39], higher-order Josephson harmonics [40]. The fields at which the effect manifests are as low as 10-20 mT, smaller than in previous works [36, 37] but large enough to dominate over possible self-field effects.

**Results list.**    In InSb-Sn nanowire junctions, we observe skewed diffraction patterns. When the magnetic field is perpendicular to the nanowire and in-plane (along the $\hat{x}$-direction), the switching current $I_c$ is inversion symmetric with respect to flux and current bias. The pattern is nearly-symmetric along the out-of-plane direction ($\hat{y}$) and along the current flow direction ($\hat{z}$) [Fig. 2(a)]. The effect is observed over wide ranges of gate voltage, which works against a fine-tuning explanation. At the same time, no consistent effect of gate voltage on the skew magnitude is observed. To interpret our results, we develop two models. The first is a phenomenological model that illustrates how a two-component CPR with $\varphi_0$ can result in a skewed diffraction pattern. The second is a numerical tight-binding model which yields the $\varphi_0$-junction.
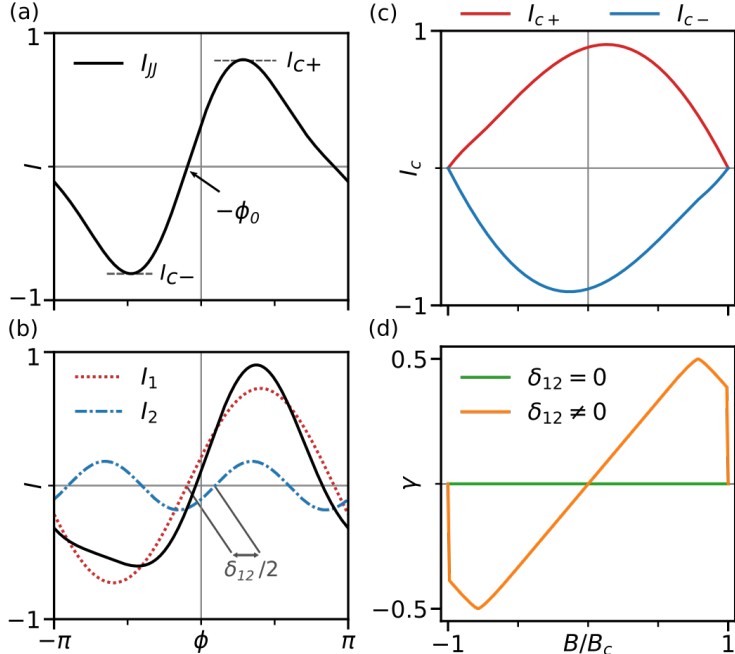

Figure 1: A phenomenological model for the skewed diffraction pattern: (a) CPR of a $\phi_0$-Josephson junctions ($\delta_{12} = 0$) at an external field $B = 0.5B_c$ with $\phi_0 = B$. Local maximum and minimum of CPR are taken as critical current flow through positive and negative bias $I_{c+}$ and $I_{c-}$. In this configuration we have $|I_{c+}| = |I_{c-}|$. (b) CPR of first (red dotted), second harmonic (blue dot-dashed), and their sum (black solid) at external field $B = 0.5B_c$. The second harmonic has a shift of ground-state phase $\delta\phi$ from the first order with a magnitude of $\delta\phi = \delta_{12}/2$. (c) Skewed diffraction pattern generated with our phenomenological model. (d) Coefficient $\gamma$ as a function of field $B_x$ when $\delta_{12} = 0$ and $\delta_{12} \neq 0$, respectively.

**Brief methods.** Junctions are prepared by coating the standing InSb nanowire with a 15 nm layer of Sn [41]. In front of the nanowire, another nanowire shadows the flux of Sn to create two disconnected Sn segments. Shadow junction wires are transferred onto chips patterned with local gates, contacts to wires are made using standard electron beam lithography and thin film deposition. Measurements are done in a dilution refrigerator with a base temperature of ∼50 mK equipped with a 3D vector magnet. Experimental panels plotting switching current are extracted from current-voltage measurements.

## 2 Phenomenological model

We present a minimal model capable of reproducing skewed diffraction patterns due to the $\varphi_0$-junction effect. This is not a microscopic model, but we also perform microscopic tight-binding calculations further below. In the phenomenological model, we postulate a CPR with the first and second sinusoidal harmonics. A recent mesoscopic model presents a possible scenario for how a phenomenological current-phase relation we discuss could arise Ref. [19]. In addition to a variable phase across the junction $\phi$, we allow for two phase offsets: the global parameter $\varphi_0$ and the relative phase offset between the first and the second harmonics, $\delta_{12}$:

$$I(\phi) = I_1 \sin(\phi + \varphi_0) + I_2 \sin(2\phi + 2\varphi_0 + \delta_{12}), \tag{1}$$

where $I_1$ and $I_2$ are the amplitude of each harmonic at zero external field.

We first explain how this CPR realizes the so-called supercurrent diode effect. If $I_2 = 0$, the CPR exhibits $I(\phi = 0) \neq 0$ with the trace shifted by $\varphi_0$ [Fig. 1(a)]. This offset can, in principle, be detected in a SQUID, but not in a single junction measurement. This is because $I_{c+} = I_{c-}$ for the maximum supercurrent in the positive and negative bias direction. The same is true if $I_2 \neq 0$, but $\delta_{12} = 0$. However, if a phase offset between the first and the second harmonics $\delta_{12} \neq 0$, we get $I_{c+} \neq I_{c-}$, and the current-voltage characteristic of the junction becomes shifted upwards or downwards [Fig. 1(b)].

This is the supercurrent diode effect. In a single junction, the phase $\phi$ is free to adjust until the maximum supercurrent is reached, detected by a switch into the finite voltage state. If the CPR has at least two components with a phase offset between them, the switching current is different for positive and negative bias directions. Note that the diode effect is not to be confused with hysteretic supercurrent in underdamped junctions.

Next we generate skewed critical $I_c$ diffraction patterns. For this we assume a phenomenological magnetic field dependence for model parameters: $I_1, I_2 \propto (1 - B^2/B_c{}^2)$, where $B_c$ is the critical field, reflecting suppression of critical current by magnetic field, and $\varphi_0, \delta_{12} \propto B$ to model the effect of spin-orbit coupling [6]. Fig. 1(c) shows the maxima in $I_{c+}$ and $I_{c-}$ located symmetrically around $B = 0$, the diffraction pattern is skewed and inversion-symmetric.

To quantify the skew, we introduce a coefficient $\gamma$, which share the same function as 'supercurrent diode coefficient/quality-factor' used in Ref [26, 29]:

$$\gamma = \frac{\Delta I_c}{(|I_c|)} = \frac{|I_{c+}| - |I_{c-}|}{(|I_{c+}| + |I_{c-}|)/2} \, . \tag{2}$$

Here $\Delta I_c$ is the difference between the magnitudes of $I_{c+}$ and $I_{c-}$, and $|I_c|$ is the average critical current. In Fig. 1(d), $\gamma = 0$ for any field when $\delta_{12} = 0$. When $\delta_{12} \neq 0$, $\gamma$ peaks at a finite field.

Experimentally $\gamma$ can also change sign as the field is increased without going through zero. The phenomenological model provides a simple explanation for this. Within the model, $\gamma = -2I_2/I_1 \sin \delta_{12}$ when $I_2 \ll I_1$. Hence, $\gamma$ changes sign when $\delta_{12} = \pi$. There is no special significance to this situation, and in particular it does not correspond to any topological transition.

## 3 Device description

A schematic of the nanowire device is depicted in Fig. 2(a). InSb nanowire (blue) covered by Sn shell (silver) is placed on top of local gate electrodes and contacted by Ti/Au (gold) contacts. The direction of supercurrent is along $\hat{z}$. The direction of spin-orbit field $B_{so}$ is induced by the breaking of inversion symmetry in the device geometry, and indicated along $-\hat{x}$ based on previous experiments [42].

The scanning electron microscope image of device A is in Fig. 2(b). The Sn shell covers 3 of the 6 facets of the hexagonal nanowire cross-section. For device A, the Sn shell faces the bottom, meaning it is oriented opposite to the schematic in Fig. 2(a). For devices B and C the shell is likely on the side (see the appendix for cross-sectional STEM and AFM imaging). In principle, the shell orientation can influence the direction of $B_{so}$, but we see no evidence of this from the measurements.

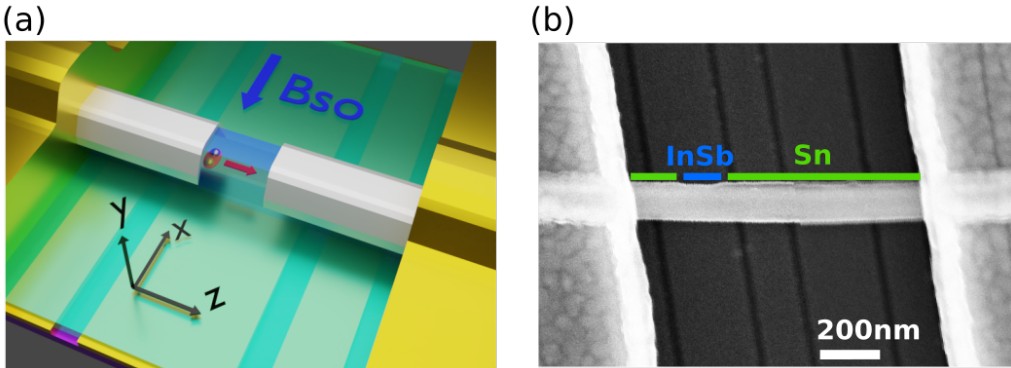

Figure 2: (a) Cartoon of a shadow nanowire Josephson Junction device. Effective spin-orbit magnetic field ($B_{so}$) is indicated. (b) Scanning Electron Microscope image of Device A.

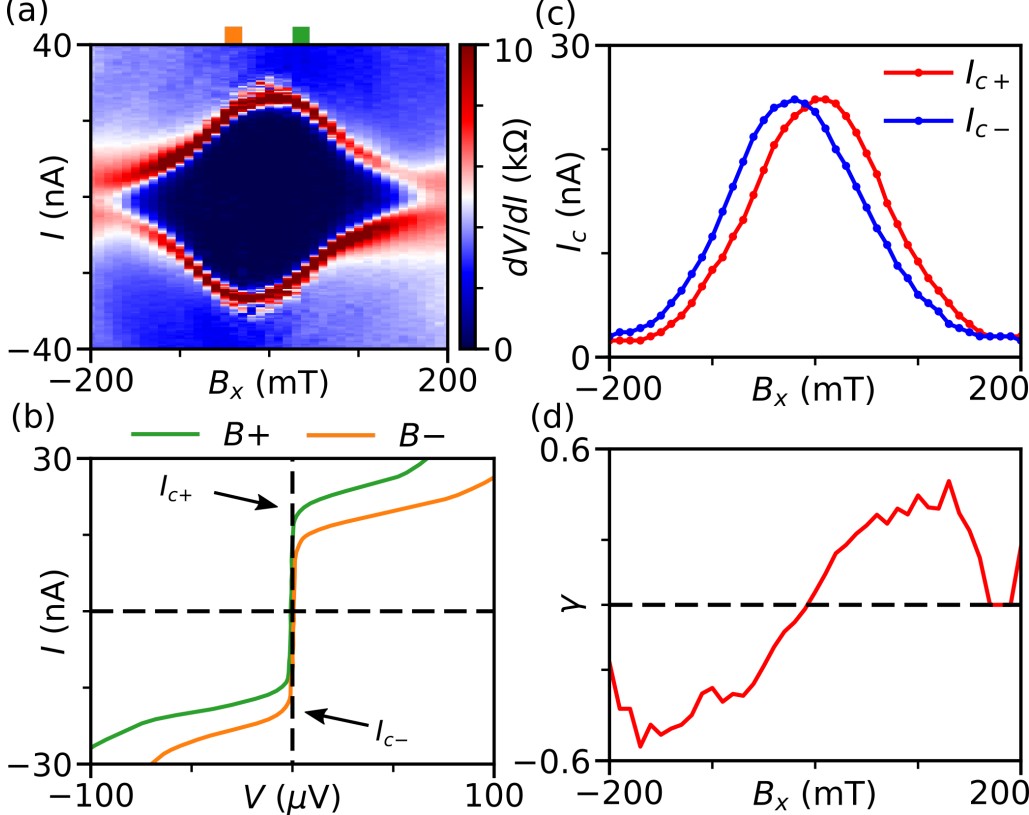

Figure 3: Skewed critical current diffraction pattern in device A. (a) At gate voltage $V_{gate} = -2V$. The normal resistance is 4 kΩ. (b) Line-cuts from (a) labeled with color squares. Critical current at positive and negative bias are labeled as $I_{c+}$ and $I_{c-}$. (c) Upper panel: switching currents extracted from panel (a), Lower panel: Coefficient $\gamma$ calculated from panel (c).

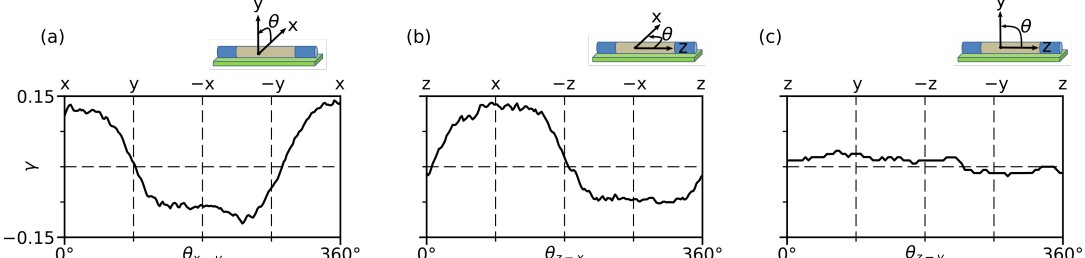

Figure 4: Field rotation in Device B. Coefficient $\gamma$ as a function of angle when the field is rotating in three orthogonal planes: x-y(a), x-z(b), and y-z(c) with fixed strength $|B| = 50$ mT. In (a) the external field is along the $\hat{y}$-axis when $\theta_{xy} = 90°$ and $270°$. In (b) the external field is along $\hat{x}$-axis when $\theta_{z-x} = 90°$ and $270°$. In (C) the external field is along $\hat{y}$-axis when $\theta_{z-y} = 90°$ and $270°$.

## 4 Skewed diffraction pattern

Fig. 3 shows a representative skewed diffraction pattern from device A. The field is applied along $\hat{x}$, in-plane and perpendicular to the nanowire. In Fig. 3(a) the switching current is where the differential resistance $dV/dI$ changes from zero (dark blue) to a finite value. The pattern is visibly inversion-symmetric. In data processing, we treat current source that gives voltage drop across the device smaller than $10\mu$V as superconducting regime and vice versa. The switching currents $I_{c+}$ and $I_{c-}$ extracted from panel (a) exhibit maxima displaced to positive and negative fields respectively [Figs. 3(b),3(c)]. By taking I-V traces at $B_x = 50$ mT and $B_x = -50$ mT, we get $\Delta I_c = 5 - 8$ nA [Fig. 3(b)], which is about 30% of $|I_c|$. The maxima of $I_{c\pm}$ are at $B_x = \pm 20$ mT. The coefficient $\gamma$ peaks at a higher field of order 100 mT right before the collapse of $I_c$ at higher fields.

Supercurrent is not hysteretic in this regime. A hysteresis would manifest in a vertical shift in the $\gamma$ dependence which would not be inversion-symmetric. We discuss data acquisition and processing in the underdamped regime in the appendix. A 10 mT hysteresis in magnet field is found in the measurements (appendix Section E) and considered in our discussion.

## 5 Magnetic field anisotropy

Device B has the same geometry as device A and its SEM picture can be found in Fig. S1(c). We measure device B in the external field $|B| = 50$ mT and rotate the field in three orthogonal planes. Critical current are traced from zero bias and extracted at where current bias gives differential resistance larger than 2 kΩ. Coefficients $\gamma$ are calculated based on extracted $I_{c+}$ and $I_{c-}$ [Fig.4(a)-(c)]. $\gamma$ in x-y and x-z planes reach zero when the external field is aligned to the $\hat{y}$ or $\hat{z}$ axis, and is large along $\hat{x}$. In the y-z plane, $\gamma$ is significantly reduced. More critical current diffraction patterns in fields at a different angle in the x-y plane can be found in Fig. S4. The junction is made with nanowires that are half-covered by superconductor shells. Shell orientation is studied in the appendix. Device A has its Sn shell on the bottom and device B has the Sn shell on the side [Fig. S2]. Nevertheless, the same general behavior, with skew coefficient $\gamma$ being largest along $\hat{x}$ is observed in devices A, B, C (see appendix C).

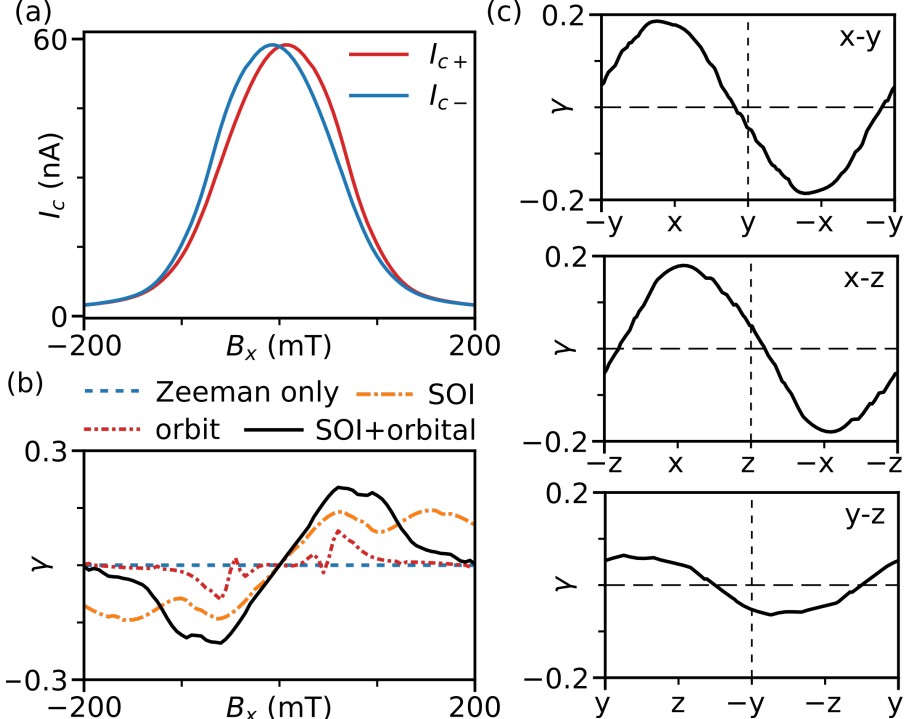

Figure 5: Numerical simulation results: (a) Critical currents ($I_{c+}$ and $I_{c-}$) as a function of magnetic field $B_x$. The chemical potential is set to two transverse or four spin-full modes ($\mu = 8$meV). (b) Coefficient $\gamma$ as a function of $B_x$ corresponding to different combinations of terms in the Hamiltonian (see legend). (c) Coefficient $\gamma$ as a function of angle $\theta$ when the external field is rotating in three orthogonal planes with fixed strength $|B| = 50$ mT. The Zeeman effect ($g = 50$) is present in all the results. Other parameters used in the simulation are $\alpha = 200$ nm $\cdot$ meV, $m_{eff} = 0.015 m_e$, temperature $T = 100$ mK. The lattice constant $a = 8$ nm, the nanowire diameter $d_1 = 120$ nm, the outer diameter (with Sn shell) $d_2 = 140$ nm and the coverage angle $\phi = 180°$. How chemical potentials were chosen in the simulation is discussed in appendix Part E.

# 6 Tight-binding model

We numerically study the microscopic properties of the system within a tight-binding model that has the same geometry as experiments (see the appendix for the description of the 3D model). This model was developed to study supercurrent interference in nanowires using KWANT [43–45]. In that project, the field orientation along the nanowire was primarily investigated. Here, we rotate the field. We can toggle on and off spin-orbit interaction ($\alpha$), the orbital vector-potential effect of external field (**A**), while Zeeman effect and disorder always remain on. For details of this model, see appendix Section G.

In Fig. 5(a) we reproduce a skewed diffraction pattern within the tight-binding model for fields oriented along $\hat{x}$. Fig. 5(b), we illustrate the role of spin-orbit interaction. The characteristic peak-antipeak structure is present whenever $\alpha \neq 0$. The coefficient $\gamma$ remains zero when only Zeeman effect of magnetic field is included.

Orbital effect only ($\alpha = 0$, $A \neq 0$) yields a similar structure (see $B_x = 80$mT). The effect appears to be weaker than the spin-orbit effect, and does not exhibit the same magnetic field anisotropy. One possibility is that this is related to the chirality-induced magnetism through the orbital effect when the geometry is lacking the inversion symmetry about the external

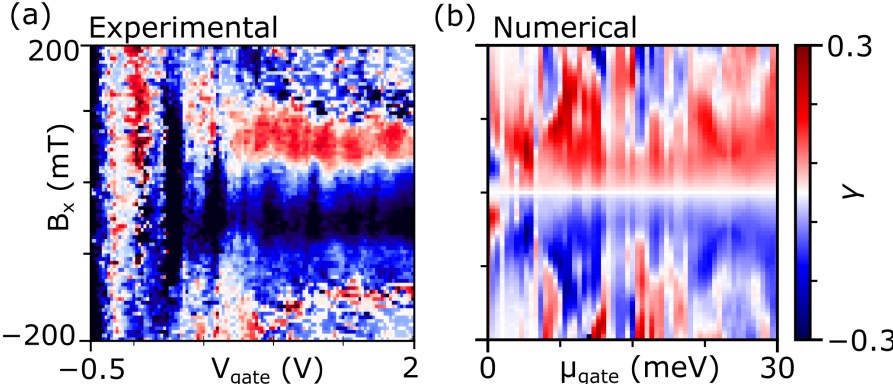

Figure 6: Comparison between the experiment (a) and the numerical simulation (b). Coefficient $\gamma$ versus external field $B_x$ and gate voltage (chemical potential $\mu$) are plotted as 2D maps to study skew shape as function of gate voltage, $V_{gate}$ ($\mu$ in simulation). The experimental data are taken from device B. Parameters used for the simulation are the same as in Fig 5. Based on the discussion in appendix Section E, the gate voltage range in our experiment is corresponding to chemical potential $\mu = 15 - 30$ meV in the simulation. Another 2D map derived from Device C can be found in Fig. 25.

field [46]. This phenomenon is of interest for a future study. See appendix Section G for detailed discussion of field direction dependence of the orbital effect. With all contributions toggled on, we reproduce the magnetic field anisotropy of coefficient $\gamma$, compare Fig. 4 and Fig. 5(c).

The tight-binding model allows us to generate current-phase relations, which exhibit $\varphi_0$ shifts as well as higher order harmonics and an extra $\delta_{12}$ emerge and increase when field is along $\hat{x}$ [Fig. 23]. This agrees with the phenomenological model of Fig. 1. The parameters used in the numerical model are discussed in the appendix. Some of the parameters, such as spin-orbit strength, exceed those previously reported for InSb nanowires [42], however we cannot claim that matching this model to data is a reliable way of extracting spin-orbit interaction strength.

# 7 Gate voltage dependence

For device B we take gate sweeps of the supercurrent in a series of external fields that are along the $\hat{x}$-axis [Fig. 6(a)]. We observe that for $V_{gate} > 0$, $\gamma$ has a characteristic double-peak shape in magnetic field which describes the skewed diffraction pattern. The behavior is observed over a significant range of gate voltages, from near pinch-off to near saturation of normal state conductance. The less regular traces at negative gate voltages are too close to pinch-off where supercurrent is small. See appendix for unprocessed gate voltage data for this and other devices.

The data demonstrate that skewed diffraction patterns do not appear only at fine-tuned values of gate voltage. On the other hand, there is no clear gate voltage dependence of the magnitude of magnetic field extent of $\gamma$. The behavior looks qualitatively similar in the tight-binding simulation [Fig. 6(b)]. Given the gate voltage range, we do not expect being able to significantly tune the bulk Rashba spin-orbit coefficient in these InSb nanowires.

In Fig. Fig. 6(a), $\gamma = 0$ (white stirpe) is reached at positive field, but not zero field as what we got in tight-binding simulation. This is an artifact of field residual in field scanning measurement, and the nature of the artifacts is explained in the appendix.

## 8 Alternative explanation: Self-field effects

Skewed diffraction patterns were observed for decades in Josephson junctions due to self-field effects, where current through the junction generates flux in the junction [47]. This effect is more pronounced in wide junctions with high critical current density. We estimate the magnetic field generated by current flow through a nanowire with the Biot-Savart Law. With current I = 300 nA, nanowire radius r = 60 nm and Junction width L = 120 nm, we get self induced field B $\sim$ 1 $\mu$T at the surface. In Fig. 2 (a) and (c), the maximum and minimum of $I_c$ are achieved at $|B| = 20$ mT, which is thousands of times larger than the self-induced field. Therefore, the self-field effect is not enough to explain the skewed pattern we observe. Furthermore, the skew due to self-fields should generally manifest for external fields along both $\hat{x}$ and $\hat{y}$, and be sensitive to the shell orientation, while the skew reported here is strongest along $\hat{x}$ for two different shell orientations.

## 9 Conclusion

Skewed diffraction patterns, observed in our experiment, are reproduced using two theoretical models. The phenomenological model, and the tight-binding model, agree that the imbalance between $I_{c+}$ and $I_{c-}$ and the critical current maxima displaced for zero field are related to the current-phase relation with two Josephson harmonics, and a phase-shift between them. The tight-binding model yields phase-shifts $\varphi_0$ and $\delta_{12}$ due to strong Rashba spin-orbit interaction. In turn, experiment shows that skew in diffraction pattern is largest when the field is oriented along $\hat{x}$, the likely orientation of spin-orbit effective field in nanowires. The effects are observed in multiple nanowires and require no fine-tuning with gate voltages.

## Acknowledgments

We thank G. Badawy, S. Gazibegovic, E. Bakkers for providing InSb nanowires. We thank Y. Nazarov, R. Mong for discussions. We acknowledge the use of shared facilities of the NSF Materials Research Science and Engineering Center (MRSEC) at the University of California Santa Barbara (DMR 1720256) and the Nanotech UCSB Nanofabrication Facility.

**Funding information** Work supported by the NSF PIRE:HYBRID OISE-1743717, NSF Quantum Foundry funded via the Q-AMASE-i program under award DMR-1906325, U.S. ONR and ARO and France ANR through Grant No. ANR-17-PIRE-0001 (HYBRID).

**Data availability** Curated library of data extending beyond what is presented in the paper, as well as simulation and data processing code are available at [48].

**Duration and volume of study** This project was started in May 2021, when the skewed diffraction pattern was first observed in nanowires Josephson junction (devices were studied longer). The experiments measurement ended in May 2022 and the simulation analysis ended in August 2022. Devices studied in this project are made of nanowires that is reported in ref [41]. 28 devices on 3 chips are measured during 5 cooldowns in dilution refrigerators, producing about 5200 datasets.

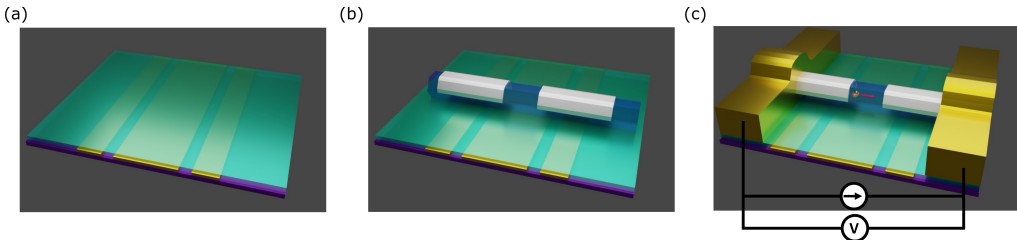

Figure 7: Fabrication steps in making nanowires Josephson junction (a) Local gate chips made on a Si substrate (purple). Ti/AuPd gates (gold) are covered by HfOx dielectric (Green) (b) InSb shadow nanowires that are half-covered by Sn shells are transferred onto the chips. (c) Metal leads made with Ti/Au are connected to the shell to form a Josephson junction.

## A   Fabrication and measurements

Nanowires are placed onto a Si chip that has predefined local gates. Electrostatic local gates are patterned by 100 keV Electron Beam Lithography (EBL) on undoped Si substrates. Local gates have mixed widths of 80 and 200nm and are separated with a distance of 40 nm. Electron beam evaporation of 1.5/6nm Ti/PdAu is used to metalize the gates which are covered by 10nm of ALD HfOx that serves as a dielectric layer [Fig. S1(a)].

Based on the Scanning electron microscope images of all devices that are studied in this report, the length of the JJs made with Sn-InSb nanowires is 120-150nm. The width of the junction is the same as the width of the nanowires, which is ≈ 120nm. After placing nanowires onto gates [Fig. 7(b)], we cover the whole chip with PMMA 950 A4 electron beam resist. Resist is dried at room temperature by pumping with a mechanical pump in a metal desiccator for 24 hours. Then we use EBL to define normal lead patterns. After development, we clean the residue of resist in an oxygen plasma asher. In the electron beam evaporator, we first use an *in-situ* ion mill to remove AlOx capping layer from the nanowires in the contact area, after which we deposit 10nm/130nm Ti/Au on the chips [Fig. 7(c)].

Transport 2-point measurements with currents source and voltage measurement model in parallel are used, with several stages of filtering placed at different temperatures.

## B   Device list and shell orientation

3 chips are studied in this project. Each chip contains multiple devices. Among these chips, 5 devices demonstrate sharp critical currents that are tuned by the gates. Skewed diffraction patterns taken from these devices are all plotted and posted in this report. In the main text, we present skewed diffraction patterns from device A [Fig. 3]. In device B, we study field direction dependence by rotating fields in different planes and study gate voltage and field effect on the skewed diffraction pattern with a 2D map [Fig. 4 and 6].

Devices A and A1 are on Chip QPC2. The EDX results show InSb nanowires and Sn shells. No ferromagnetic materials were found in the junction. Sn shells are half-covering the nanowires from the bottom side and are in touch with the HfOx dielectric layer.

Device B is on Chip QPC4. The shell orientation of the device is studied with Atomic Force Microscopy [Fig. S2(c)]. Based on the AFM images, we conclude that the Sn shell is on the side.

Devices C and C1 are on Chip QPC3. The shell orientations are not studied.

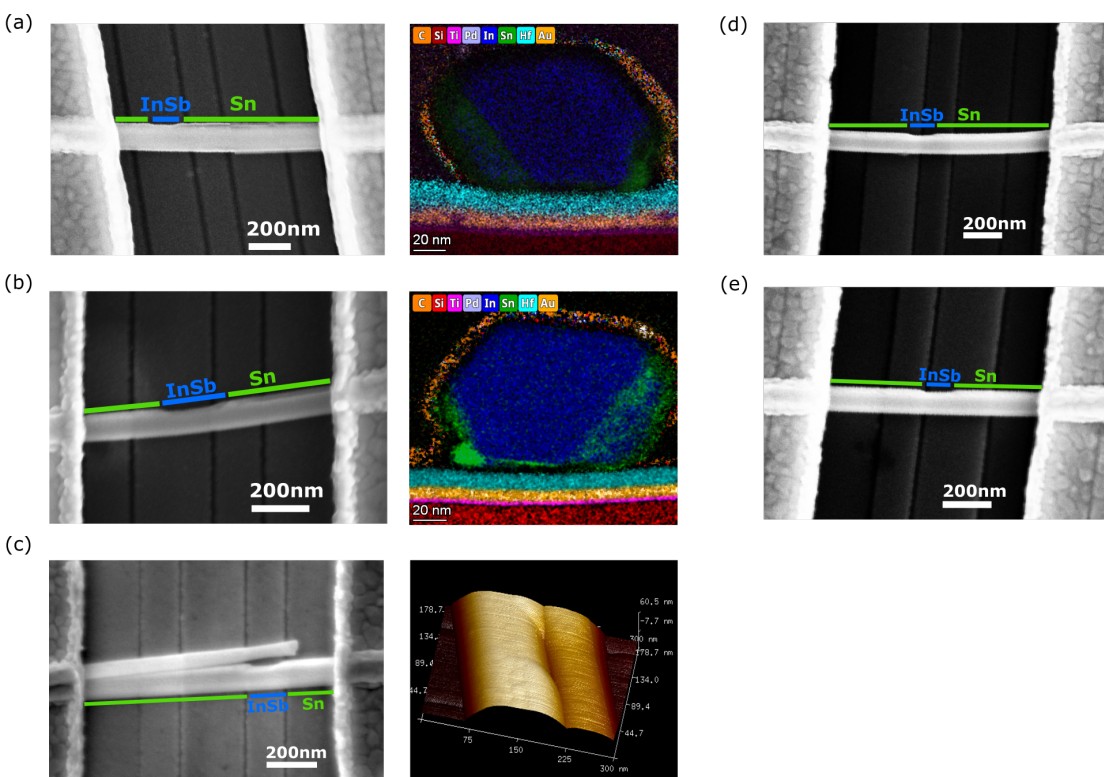

Figure 8: Scanning electron microscope (SEM) image of Devices measured in report. (a) Right: SEM image of Device A (Chip: QPC2). Left: Scanning transmission electron microscopy (STEM) based Energy-dispersive x-ray spectroscopy (EDS) spectroscopic elemental image of Device A. (b) Right: SEM image of Device A1 (Chip: QPC2). Left: STEM based EDS spectroscopic elemental image of Device A1. (c) Right: SEM image of Device B (Chip: QPC4). Left: Atomic force microscopy (AFM) image of Device B. (d) SEM image of Device C(Chip: QPC3) (e) SEM image of Device C1 (Chip: QPC3).

# C  Diffraction pattern at different gates in field along three axes

## C.1  Device A



Figure 9: Diffraction patterns at different gates in the field along three axes, measured in Device A (QPC2). (a) dV/dI differential resistance as a function of $\hat{x}$-direction field $B_x$ and current bias. $\gamma$ is calculated with the extracted magnitude $\Delta I_c$. (b) Diffraction pattern when the field is applied parallel to the nanowires, along $\hat{z}$-direction. (c) Diffraction pattern when the field is applied out of substrate plane, along $\hat{y}$-direction.

## C.2 Device A1

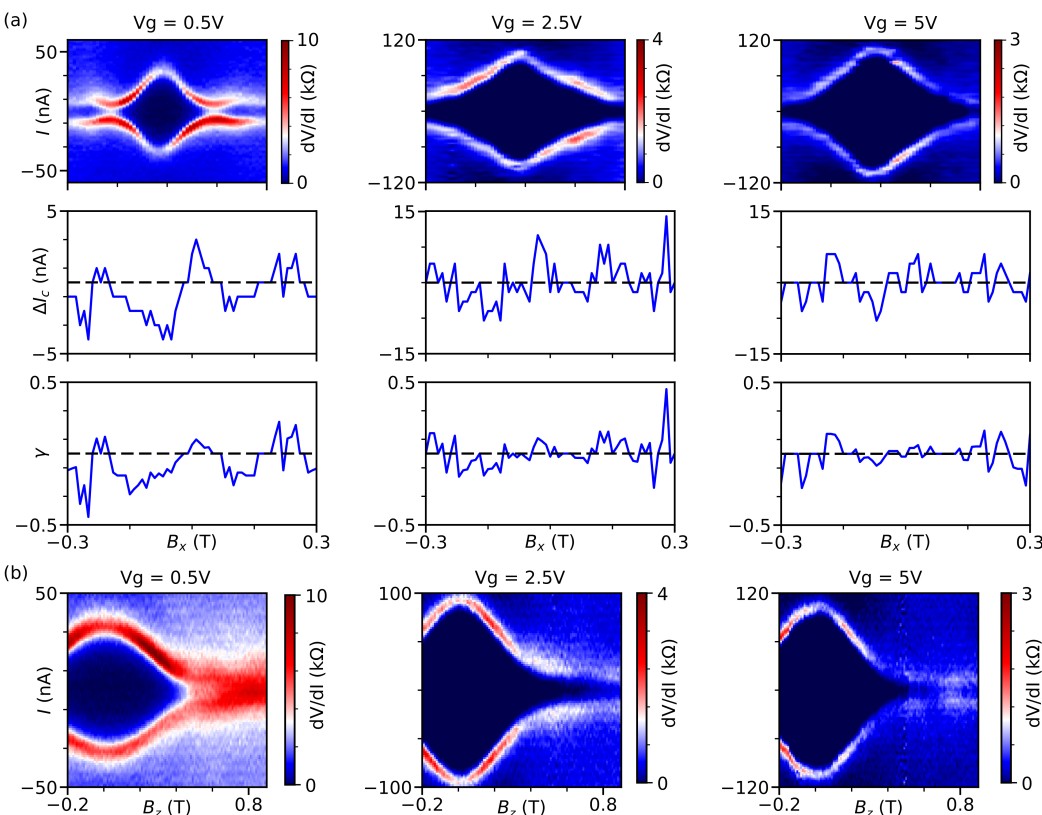

Figure 10: Diffraction patterns at different gates in the field along $\hat{x}$ and $\hat{z}$ axes, measured in Device A1 (QPC2). (a) dV/dI differential resistance as function of $\hat{x}$-direction field $B_x$ and current bias. $\gamma$ is calculated with the extracted magnitude $\Delta I_c$. (b) Diffraction pattern when the field is applied parallel to the nanowires, along $\hat{z}$-direction.

## C.3   Device B

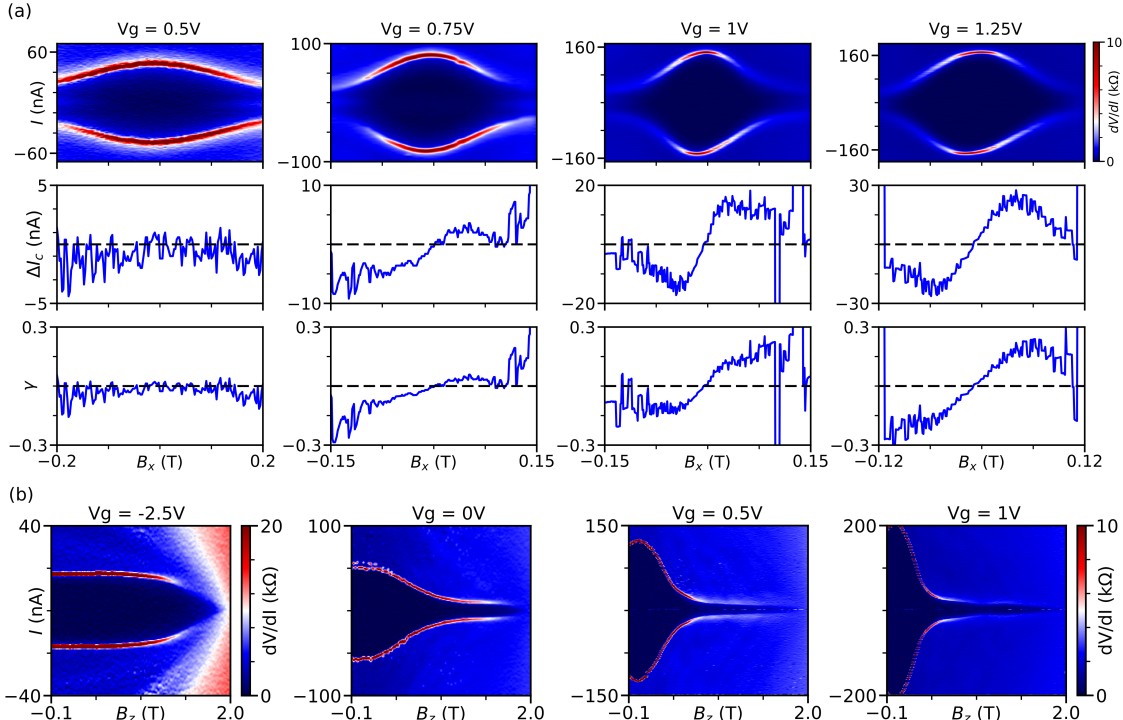

Figure 11:   Diffraction patterns at different gates in the field along $\hat{x}$ and $\hat{z}$ axes, measured in Device B (QPC4).  (a) dV/dI differential resistance as function of $\hat{x}$-direction field $B_x$ and current bias. $\gamma$ is calculated with the extracted magnitude $\Delta I_c$. (b) Diffraction pattern when the field is applied parallel to the nanowires, along $\hat{z}$-direction.

## C.4 Device C

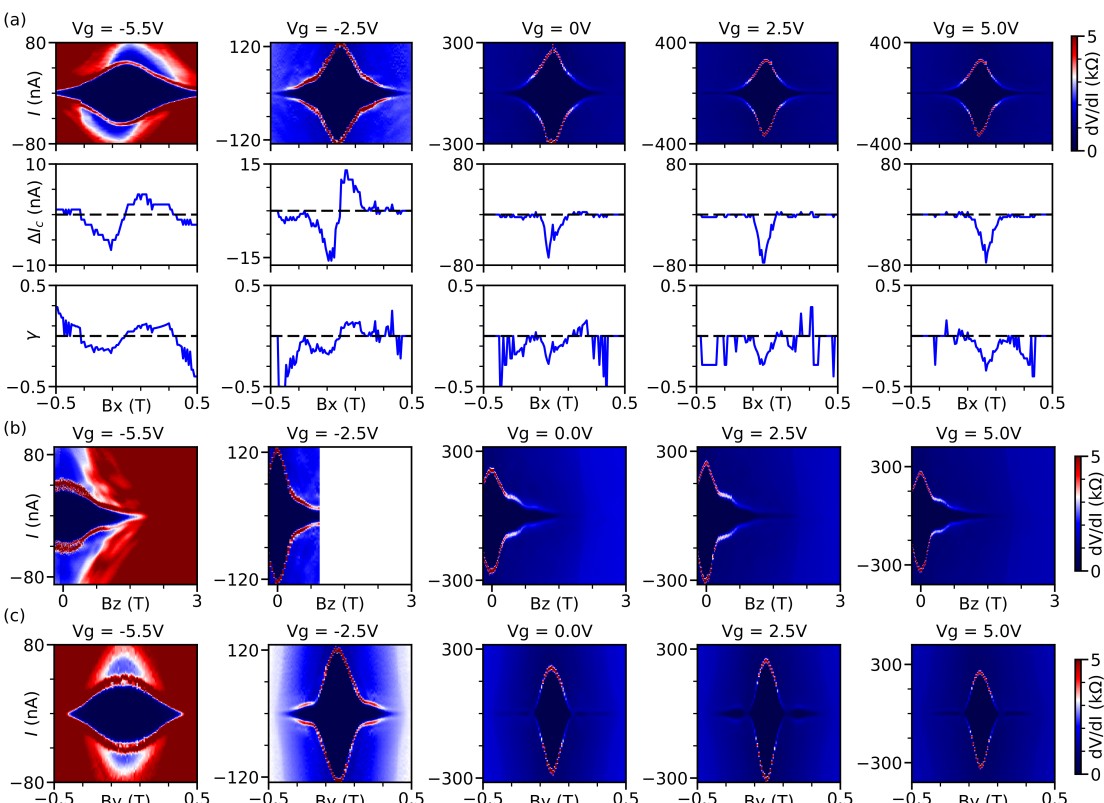

Figure 12: Diffraction patterns at different gates in the field along three axes, measured in Device C (QPC3)(a) dV/dI differential resistance as function of $\hat{x}$-direction field $B_x$ and current bias. $\gamma$ is calculated with the extracted magnitude $\Delta I_c$. (b) Diffraction pattern when field is applied parallel to the nanowires, along $\hat{z}$-direction. (c) Diffraction pattern when the field is applied out of substrate plane, along $\hat{y}$-direction.

## C.5 Device C1

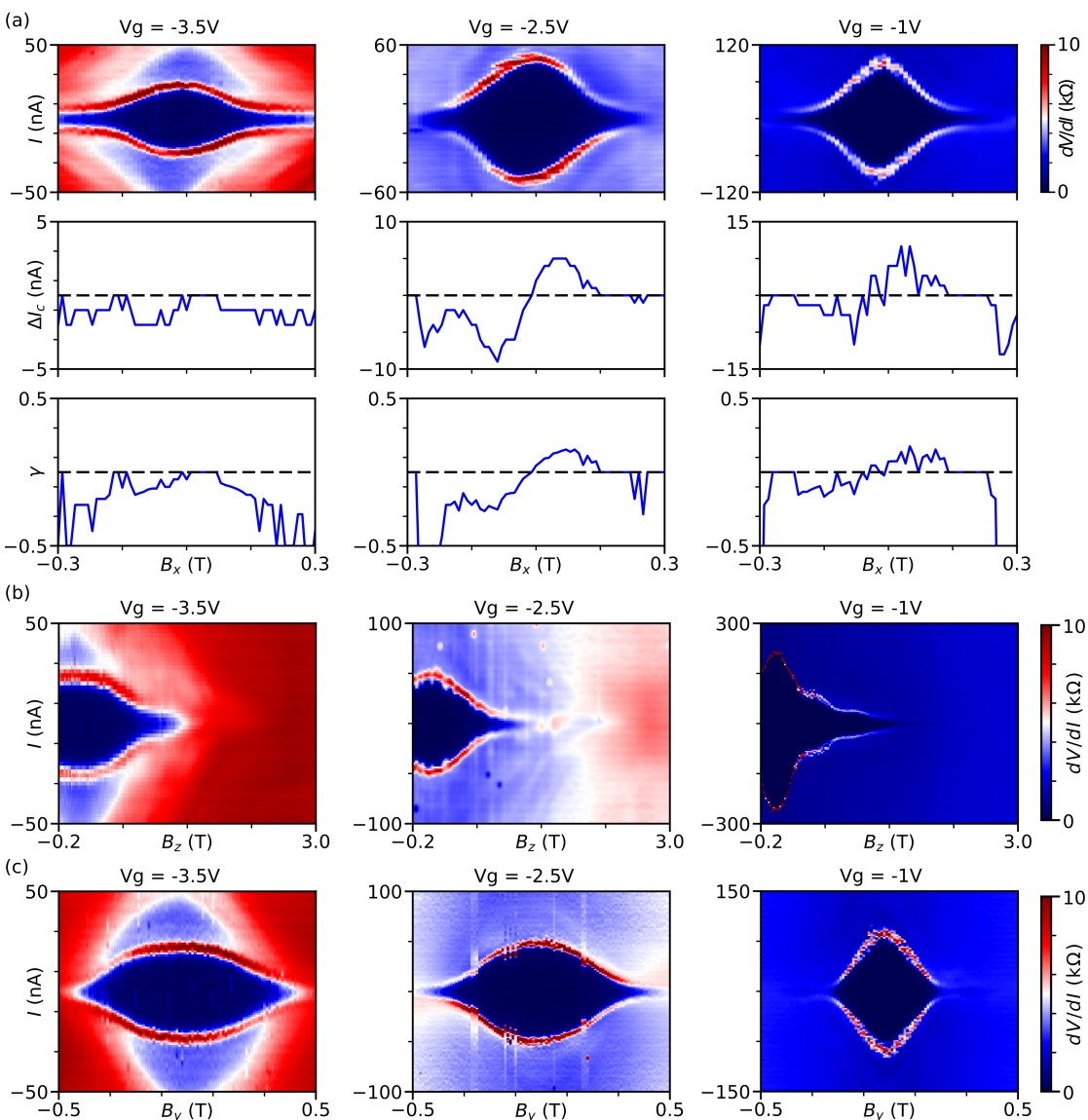

Figure 13: Diffraction patterns at different gates in the field along three axes, measured in Device C1 (QPC3) (a) dV/dI differential resistance as function of $\hat{x}$-direction field $B_x$ and current bias. $\gamma$ is calculated with the extracted magnitude $\Delta I_c$. There is a shift of gate, so strength of $I_c$ may be different from scans with other two fields directions. (b) Diffraction pattern when field is applied parallel to the nanowires, along $\hat{z}$-direction. (c) Diffraction pattern when the field is applied out of substrate plane, along $\hat{y}$-direction.

# D Diffraction pattern at x-y plane, field is applied along angle $\theta$ between 0° to 180°, Device C

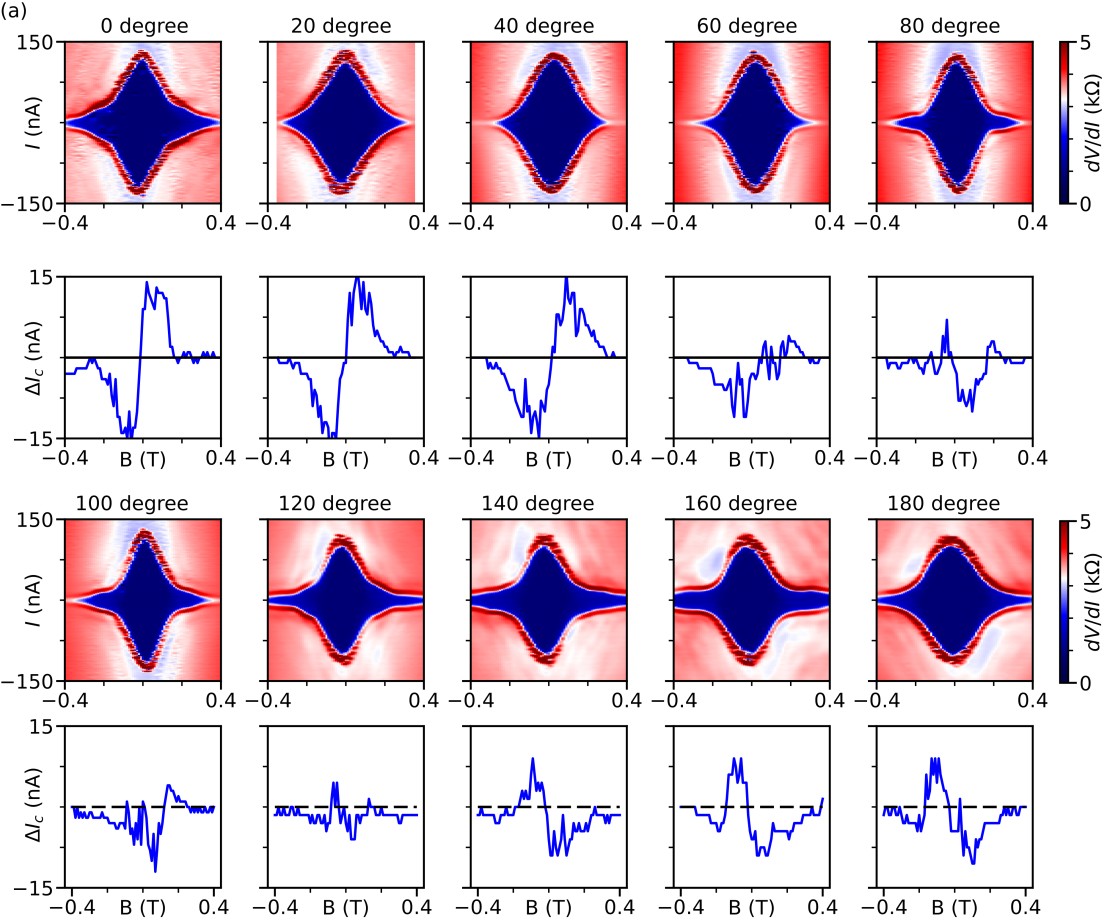

Figure 14: Diffraction pattern at the x-y plane, the field is applied along angle $\theta_{x-y}$ between 0° to 180°, measured in Device C. Critical current difference $\Delta I_c$ is extracted from measurement data using a peak finder Python script.

# E Characteristics of the junctions and hysteresis in the measurement setup

The gate effect is studied by applying a bias voltage across the device $V_{bias} = 10mV$. Conduction channels in Device A [Fig. S9(a)] can be fully closed by the tunnel gate. While Devices B and C [Fig. S9(d),(g)] cannot be fully closed. The Josephson effect at zero magnetic field is best studied in the current-bias configuration [Fig. S9(c),(f),(i)]. The switching current from superconducting to normal state regime is demonstrated with a read peak in differential resistance (referred to in other Figures as $I_c$). The magnitude of $I_c$ is calculated with $|I_c| = (|I_{c+}| + |I_{c-}|)/2$, where $I_{c+}$ and $I_{c-}$ are extracted with a peak finder Python script by finding the two of largest differential resistance at each gate voltage. $I_c$ increases at more positive gate voltage, while the extracted products $I_c R_N$ ($R_N$ is the normal state resistance) are in the range of 200-300 $\mu \cdot eV$.

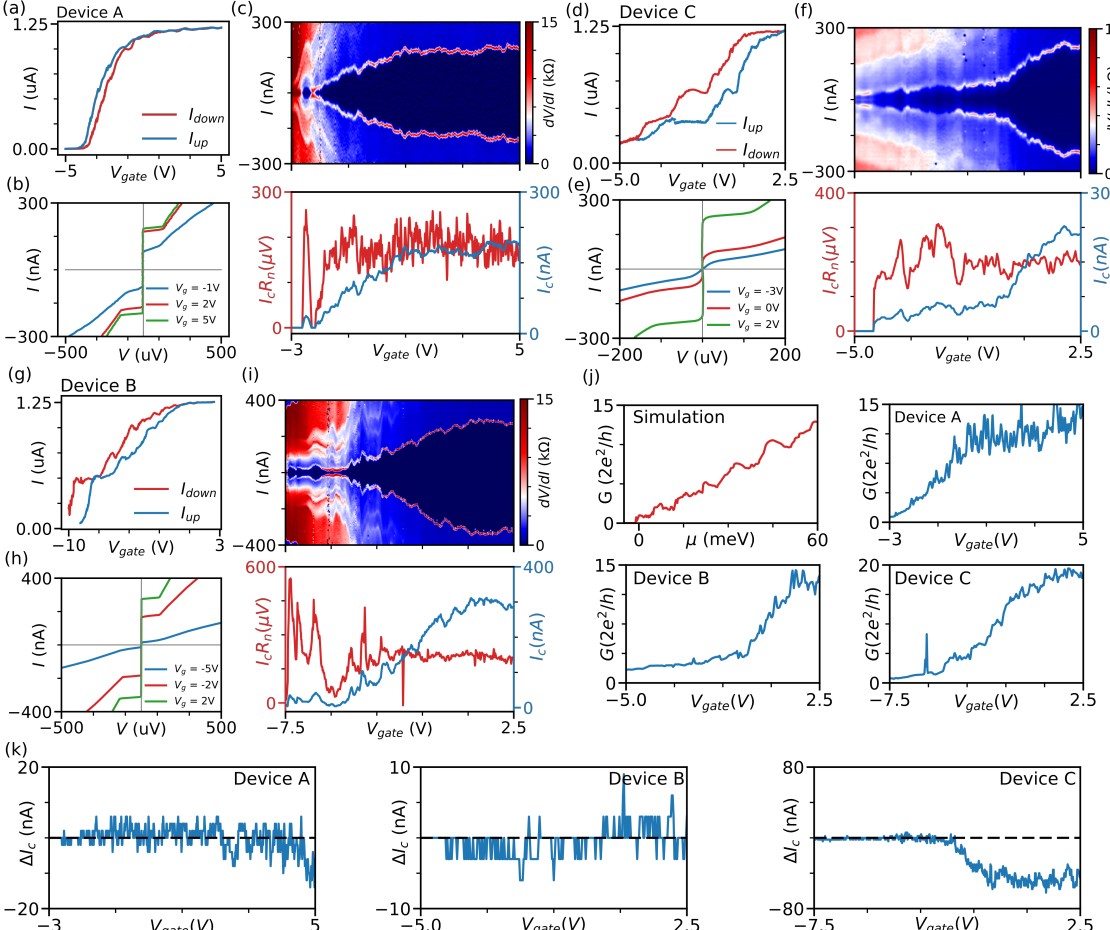

Figure 15: Gate dependence taken at external field $|B| = 0$, measurement data are taken from Device A (a-c), Device B (d-f) and Device C (g-i). (a,d,g) Current through the JJs as function of gate voltage $V_{gate}$ when bias voltage is set to be $V_{bias} = 10mV$. (b,e,h) V-I characteristics taken at different gate voltages. (c,f,i) Upper panel: dV/dI differential resistance as a function of current source and $V_{gate}$. Lower panel: extracted critical current $I_c$ (blue) and $I_c R_N$ product (red) as functions of $V_{gate}$. (j) Electrical conductance of the junction in the unit of quantum conductance ($2e^2/h$) as a function of chemical potential $\mu$ when simulating with parameters used in other figures. Followed by normal state conductance as a function of gate voltage $V_{gate}$ measured in device A, B, and C, respectively. Differential resistance are extracted from (c,f,i) and converted to conductance for plotting. (k) Critical current difference calculated with $\Delta I_c$ as functions of gate voltage at zero field, measured in device A, B, and C. $I_{c+}$ and $I_{c-}$ are extracted from (c,f,i).

## E.1 Chemical potential used in simulation and corresponding gate voltage in experimental measurements

In an attempt to establish additional correspondence between experiment and theory, we study conductance as a function of chemical potential $\mu$ in the simulation [Fig. S9(j)]. The normal state resistance read from skewed pattern in Fig. 2(a) in main text is 4 kOhm, which is close to two transverse modes or four spin-full modes. So we choose $\mu = 8$ meV to study the skew shape in Fig. 4. When studying the direction-dependent supercurrent transport as function of field direction, we get the normal state resistance about 1.5-2 kOhm. Which is corresponding

to six to eight transverse modes or twelve to eighteen spin-full mode, and chemical potential $\mu \approx 20 meV$ in the simulation.

The skew map presented in Fig. 5 in main text is measured from Device B. By comparing normal state conductance $G$ as a function of $V_{gate}$ in simulation and measurement data [Fig. S9(j)], we conclude the range used in Fig. 5, which is -0.5V to 2V, is corresponding to chemical potential $\mu = 15 - 60 meV$. Another skew map measured with Device C in Fig. S18 (c) has gate voltage range from -3V to 2V, which is corresponding to $\mu = 30 - 60 meV$.

### E.2 Hysteresis in the Josephson junction

Hysteresis in the current-voltage characteristics is studied by extracting $\Delta I_c = |I_{c+}| - |I_{c-}|$ from devices A, B, and C and plotting them as a function of gate voltage at zero magnetic fields [Fig. S9(k)]. We find the magnitude of $I_{c-}$ is larger than $I_{c+}$ at more positive gate voltage in Devices A and C. When extracting $\Delta I_c$ from skewed diffraction patterns [Figs. 9(b), 12(c)], it is also easy to see the critical current is larger at negative bias, which results in the gamma as a function of the $B_x$ field that is no longer inversion symmetric about zero fields, zero current. While in Device B, we didn't see the obvious difference between $I_{c+}$ and $I_{c-}$.

There are two methods used in current configuration measurements: 1. Unidirectional current sweeps, either from positive to negative, or from negative to positive bias. This method is used in Devices A and C. Which results in the hysteresis in current bias.

2. Sweep from zero bias. At a fixed gate voltage or field, scan from zero current bias to the positive and negative side respectively (0 to $I_+$ and 0 to $I_-$). So one data set only records scans with $I$ larger than zero and vice versa. Then we combine two datasets to get a full scan. By doing this, we can get rid of the hysteresis because only switching currents are measured. This method is used in device B and the results are plotted in [Fig. 4, 6].

In the main text, the skewed diffraction pattern from Device A are studied at a smaller gate voltage, $V_{gate} = -2V$ [Fig. 3]. At this gate voltage, hysteresis in the junction is not observed.

### E.3 Hysteresis in the superconducting magnet

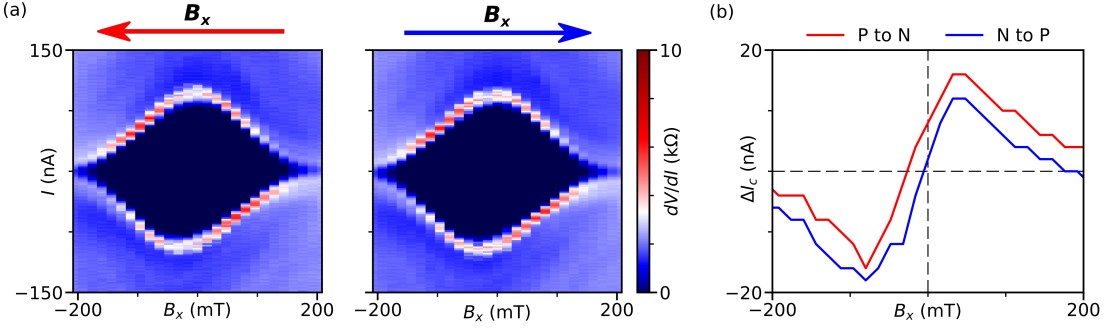

Figure 16: (a) Skewed critical current diffraction patterns, field is scanned from positive to negative and from negative to positive direction in adjacent panels. Device A is used in this scan and gate voltage is set to be $V_{gate} = $ -1V for this scan. (b) Extracted critical current difference $\Delta I_c$ is plotted as a function of $\hat{x}$-field from two scans in (a).

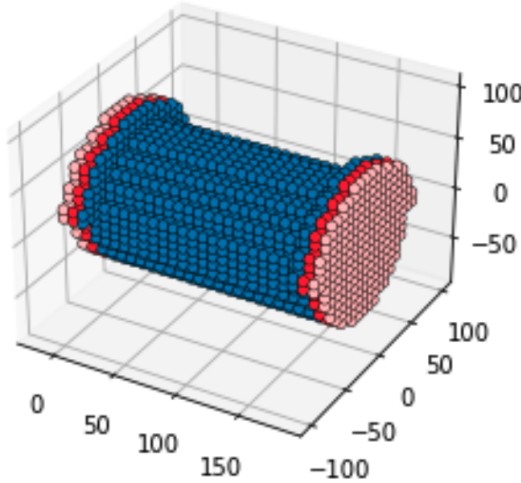

**Figure 17:** Tight-binding model generated by the function *kwant.continuum.discretize*. Red dots represent the infinite leads.

## F  Model used in simulation

To simulate the superconductor-nanowire-superconductor Josephson junction in the presence of external magnetic field, we consider the following Hamiltonian for a nanowire that is covered by superconductor lead at both ends.

$$H = \left( \frac{\mathbf{p}^2}{2m^*} - \mu + \delta U \right)\tau_z + \alpha(p_z\sigma_x - p_x\sigma_z)\tau_z + g\mu_B\mathbf{B}\cdot\hat{\sigma} + \Delta\tau_x, \tag{F.1}$$

where $\tau_i$ and $\sigma_i$ are Pauli matrices act on particle-hole and spin space respectively. $\mathbf{p} = -i\hbar\nabla + e\mathbf{A}\tau_z$ is the canonical momentum, and the magnetic potential $\mathbf{A}$ is chosen to be $[0, B_z x - B_x z, -B_y x]$, so that it is invariant along the $x$−direction. Further, $m^*$ is the effective mass, $\mu$ is the chemical potential and $\delta U$ represent the onsite disorder inside the nanowires. The Zeeman effect is given by $g\mu_B\mathbf{B}\cdot\hat{\sigma}$ and the Rashba spin-orbit coupling is given by $\alpha(p_z\sigma_x - p_x\sigma_z)$. Finally, $\Delta$ is the superconducting pairing potential.

We first construct a tight-binding model based on the Hamiltonian (F.1), then the critical current under different parameter configurations can be obtained from the imaginary part of the Green's function. These Green's functions can be obtained by using the KWANT package. We explain the code in detail below.

The first step is to construct a system with the scattering region and leads. Here we use the function *kwant.continuum.discretize* to convert the 3D translational symmetric Hamiltonian (F.2) into a tight-binding system (Figure 17).

$$H = \left( \frac{\hbar^2(p_x^2 + p_y^2 + p_z^2)}{2m^*} - \mu + \delta U \right)\tau_z + \alpha(p_z\sigma_x - p_x\sigma_z)\tau_z + g\mu_B\mathbf{B}\cdot\hat{\sigma} + \Delta\tau_x. \tag{F.2}$$

Here the Hamiltonian does not contain the orbital effect because *kwant.continuum.discretize* cannot handle the systems with lower symmetry. To include the orbital effect, we need to apply the Peierls substitution to the hopping term. The hopping between two sites $\vec{x}$ and $\vec{x}_0$ becomes $t \to te^{i\phi}$, where $\phi = -e\mathbf{A}\cdot(\vec{x} - \vec{x}_0)/\hbar$.

In order to calculate the critical current, besides normal leads and superconducting leads (red region in Fig. 17), we need to add a virtual self-energy lead to this system. Here we attach

the lead in the middle of the nanowire (yellow region in Fig. 18). Notice this self-energy lead is not connected to external devices, and is only used to calculate the Green's function. Usually, the Green's function of an infinite system contains infinite entries. But now we can divide this nanowire into two parts: self energy lead (L) and the other region (R). Then, we have

$$\begin{pmatrix} G_L^r & G_{LR}^r \\ G_{RL}^r & G_R^r \end{pmatrix} = \begin{pmatrix} E + i\eta - H_{Lead} & H_C \\ H_C^\dagger & E + i\eta - H_R \end{pmatrix}^{-1}, \tag{F.3}$$

where $H_C$ and $H_C^\dagger$ are the hopping between the lead and the rest of the nanowire. By solving this equation we get

$$G_L^r = \left( E - H_{Lead} - H_C^\dagger (E - H_R)^{-1} H_C^\dagger \right)^{-1}. \tag{F.4}$$

Thus the Green's function of the finite self energy lead contains the information about the whole system.

In the KWANT package, the retarded Green's function of the self-energy lead can be obtained by using the function *kwant.solvers.greens_function*. We first calculate the Green's function $G_L^r(0)$ without the phase difference between the two superconducting leads. Then the Green's function with the phase difference $\varphi$ can be obtained by modifying the Hamiltonian $H_{Lead}$ in equation (F.4). To be more precise, we change the hopping term $t$ to $te^{i\varphi}$, where $t$ is the hopping from the left side of the self-energy lead to the right side.

Critical current under finite temperature $T$ can be calculated by using the imaginary Green's function. Consider the self-energy lead as a subsystem. The current equals the change in the number of electron on the left side of the lead.

$$I = ie \left\langle \sum_{i \in L} \frac{dn_i}{d\tau} \right\rangle = \frac{ie}{\hbar} \left\langle \sum_{i \in L} [c_i^\dagger(\tau) c_i(\tau), H_{Lead}] \right\rangle. \tag{F.5}$$

Here we consider the imaginary time evolution and $i$ runs through the positions to the left of the self-energy lead. For any diagonal term $c_j^\dagger c_j$ in $H_{Lead}$, we have

$$[c_i^\dagger c_i, c_j^\dagger c_j] = 0. \tag{F.6}$$

For $j, k \neq i$, we have

$$[c_i^\dagger c_i, c_j c_k] = [c_i^\dagger c_i, c_j^\dagger c_k] = [c_i^\dagger c_i, c_j c_k^\dagger] = 0. \tag{F.7}$$

Therefore, to have the non-zero commutator, we need at least one operator $c_j$ or $c_j^\dagger$ such that $j \in L$. Suppose $j, k \in L$, then we have

$$[c_j^\dagger c_j, c_j^\dagger c_k] = c_j^\dagger c_j c_j^\dagger c_k - c_j^\dagger c_k c_j^\dagger c_j = c_j c_k - 0 = c_j^\dagger c_k, \tag{F.8}$$

$$[c_k^\dagger c_k, c_j^\dagger c_k] = c_k^\dagger c_k c_j^\dagger c_k - c_j^\dagger c_k c_k^\dagger c_k = 0 - c_j c_k = -c_j^\dagger c_k. \tag{F.9}$$

These two term cancel each other, thus only hopping between the left side and right side of the lead contribute to the critical current. Then the equation (F.5) simplifies to

$$I = \frac{ie}{\hbar} \left\langle \sum_{i \in L, j \in R} [c_i^\dagger(\tau) c_i(\tau), t_{ji} c_i^\dagger(\tau) c_j(\tau) - t_{ij} c_j^\dagger(\tau) c_i(\tau)] \right\rangle \tag{F.10}$$

$$= \frac{ie}{\hbar} \sum_{i \in L, j \in R} (t_{ji} \langle c_i^\dagger(\tau) c_j(\tau) \rangle - t_{ij} \langle c_j^\dagger(\tau) c_i \rangle(\tau)). \tag{F.11}$$

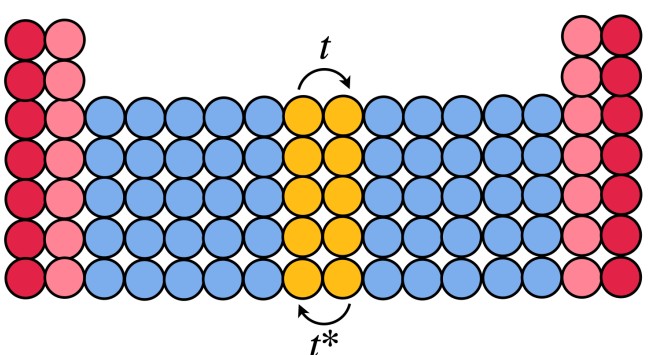

Figure 18: Cross section of the nanowire. Here we add a two-layer self energy lead in the middle of the wire, where $t$ is the hopping term from the left side of self energy lead to the right side.

By using the definition $G(\tau, \tau')_{ij} = \langle\langle c_i^\dagger(\tau) c_j(\tau')\rangle$ for $\tau > \tau'$ we have

$$I = \frac{ie}{\hbar} \sum_{i \in L, j \in R} \left( t_{ij} G(\tau, \tau')_{ij} - t_{ji} G(\tau, \tau')_{ji} \right). \tag{F.12}$$

Then by apply the inverse Fourier transformation on the right hand side of this equation and take the limit $\tau - \tau' \to 0^+$, we get

$$I = \frac{ie}{\hbar} \sum_{i \in L, j \in R} \sum_{n \in \mathbb{Z}} k_B T \left( t_{ji} e^{i\omega_n(\tau-\tau')} G(i\omega_n)_{ij} - t_{ij} e^{i\omega_n(\tau-\tau')} G(i\omega_n)_{ij} \right) \tag{F.13}$$

$$= \frac{iek_B T}{\hbar} \sum_{n \in \mathbb{Z}} \sum_{i \in L, j \in R} \left( t_{ji} G(i\omega_n)_{ij} - t_{ij} G(i\omega_n)_{ij} \right) \tag{F.14}$$

$$= \frac{-4ek_B T}{\hbar} \sum_{n \in \mathbb{N}} \text{Im}\{\text{Tr}(T_{RL} G(i\omega_n)_{LR} - T_{LR} G(i\omega_n)_{RL})\}, \tag{F.15}$$

where $T_{LR}$ and $T_{RL}$ are the hopping matrices from left (right) to right (left), $\omega_n = (2n+1)\pi k_B T$ is the $n$-th Matsubara frequency for electron. Factor 4 on the last line comes from positive-negative symmetry when sum over all the integers and particle-hole symmetry of the system.

# G  Parameters used in the simulations

In this section, we discuss the strength of parameters used in the simulation that give the best match to experimental data.

## G.1  Temperature

In the measurements, lattice temperature can only be estimated by reading the temperature from the sensor on the dilution refrigerator mixing chamber plates and it is varying from 50 to 60 mK while scanning the external field. However, electron temperature is the relevant parameter for the simulation. In our case, electrons are cooled down from room temperature to base temperature by several stages of filters, especially the cooper powder filter. However, the temperature of electrons is usually a bit higher than the device temperature. So we simulate

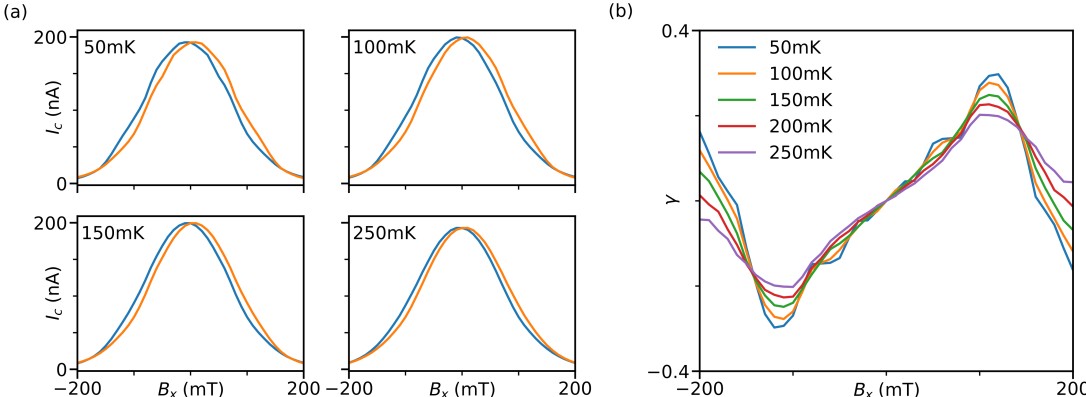

Figure 19: (a) Skewed diffraction pattern from simulation when temperature $T = 50$mK, $100$mK, $150$mK and $250$mK, respectively. (b) Coefficient $\gamma$ as function of $B_x$ at different temperatures.

the skewed shape in different temperatures (Fig.S13) and choose 100mK for all the simulation results presented in the main text.

From the simulation results, we also find that the maximum and minimum value of coefficient $\gamma$ become smaller at higher temperature. This is consistent with our temperature dependence measurement results [Fig. 24].

## G.2 Strength of spin-orbit interaction

The strength of Spin-orbit coupling is estimated by studying the critical current diffraction pattern and choosing the one which best reproduced the experiment results. We set chemical potential $\mu = 8$ meV, which is corresponding to two transverse or four spin-full modes, same value is used in the [FIg. 5] in the main text. The skew shape we observe experimentally has two features that we aim to reproduce: 1) the largest critical current $I_c$ is not located at zero field; 2) the largest critical current difference is around $B_x = \pm 50 mT$. Based on the skewed shape simulated with different strengths of spin-orbit coupling, we find the skew is best reproduced with $\alpha = 200 nm \cdot meV$. So we choose $\alpha = 200$ for all simulation results in the main text. Note that the true strength of spin-orbit interaction in nanowires may differ from the parameters in a tight-binding simulation.

## G.3 Skew shape and field rotation simulation at another chemical potential

In the main text we present skewed diffraction pattern from Device A and rotation from Device B (field rotation was not performed for device A). Thus simulation at two separate chemical potential should be considered in preparing this report. To maintain consistency of simulation, we choose $\mu = 8$meV for the simulation in the main text, Fig. 4. Here we show simulation results at $\mu = 20$meV here.

## G.4 Orbital induced skewed diffraction patterns

To study the orbital induced skewed diffraction, we set spin-orbital strength $\alpha = 0$ and $\mu = 10$ for all the simulation in Fig. 22. The shell is set to be on the top of the nanowires and have zero relative angle about the normal direction of the substrate. From the simulation, we find when the device geometry is inversion symmetry about the external field direction, there is no skewed diffraction pattern about the field. When the device geometry is not inversion symmetry about the external field, there is skewed diffraction pattern. This is the origin of the

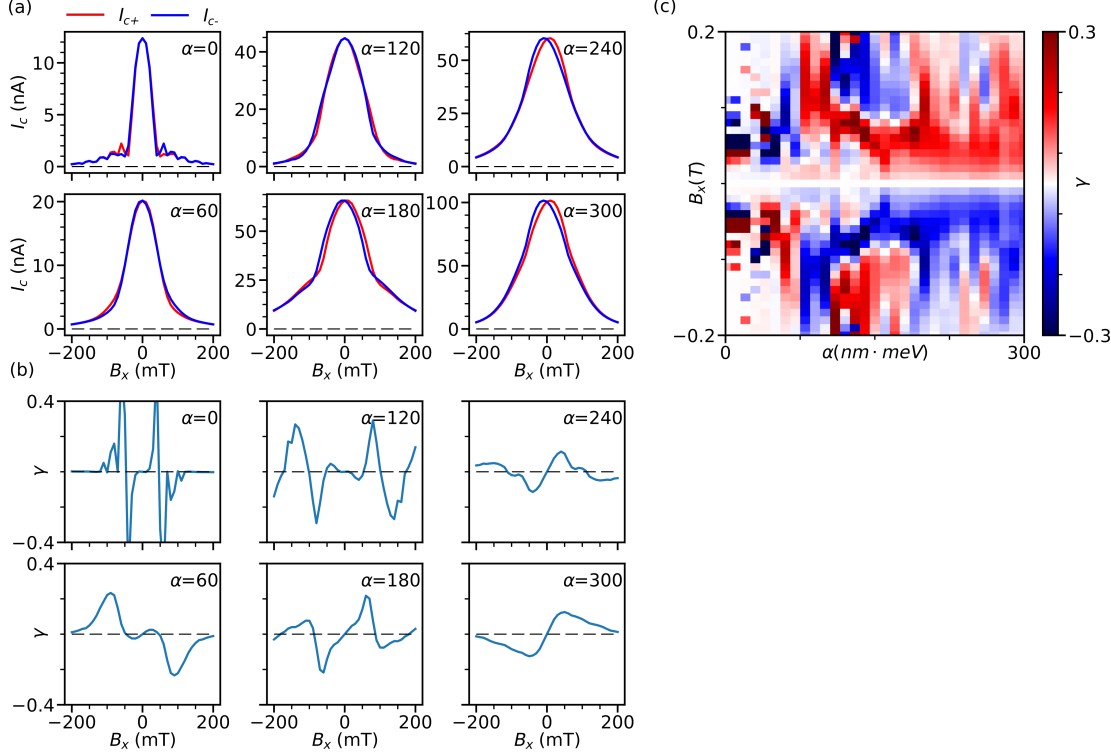

Figure 20: (a)Critical current as function of $\hat{x}$-field when strength of spin-orbit interaction $\alpha = 0, 60, 120, 180, 240, 300 nm \cdot meV$. The chemical potential $\mu = 8meV$. (b) Coefficient $\gamma$ are extracted from (a) and plotted as function of $B_x$ simulated with different $\alpha$. (c)At same chemical potential, plot coefficient $\gamma$ versus perpendicular field $B_x$ and spin-orbit strength $\alpha$ as a 2D map to study how alpha affect skew shape.

non-zero $\gamma$ in Fig. 5 (b) when only orbital effect is included in simulation. By combining this result and the skewed shape we got at $\mu = 20$ in Fig. 21(c), it is indicating the orbital effect is related to the chemical potential.

However, such effect in nanowires system cannot be fully understand through our current simulation and need further exploration. So we will leave the discussion here. More detailed studies about this topic will be presented with nanowires devices that lacking inversion symmetry by placing shells at different facets of the nanowires.

# H  Current phase relation derived from simulation results

The current phase relation (CPR) is studied within the model and its parameters suggested by comparison with experiment. We plot the CPR curve when the strength of the external field is 0T, 0.05T, 0.1T in $B_x$, $B_y$ and $B_z$ directions [Fig. S16]. Parameters used in this simulation is same as that in Fig.4 in main text. In Fig.S16(a), we find that only when the external field is along $\hat{x}$-direction, there is a shift of the ground state phase in CPR. The numerical CPR curves are similar to those postulated in the phenomenological model (Eq. 1 and Fig. 1).

To study how the time-reversal symmetry is broken when external field is along $\hat{x}$-direction. We first perform a Fourier expansion with the simulated CPR [Fig.S16 (b)]. We find there is a significant second order sin term in the simulation. What's more interesting is the first and second order cos terms are also large compared to the sin term and they are first increasing when field is applied but decreasing at higher field. This explains why the phase of ground

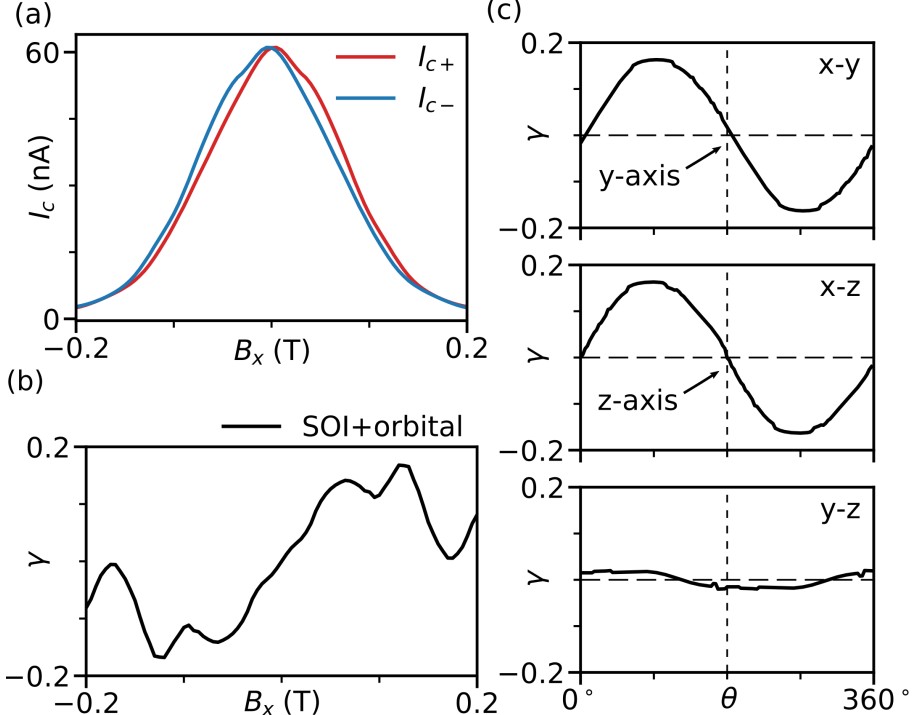

Figure 21: Numerical results from the KWANT simulation with same parameters used in Fig. 5 but with another chemical potential $\mu = 20meV$. This chemical potential is corresponding to four transverse or eight spin-full modes. (a) Critical current flow through each polarization as a function of magnetic field $B_x$. (b) Coefficient $\gamma$ as a function of $B_x$, the magnitude of $I_c$ is derived from (a). (c) Coefficient $\gamma$ as a function of angle $\theta$ when the external field is rotating in three orthogonal planes with fixed strength $|B| = 50$mT.

state shift by such a large value. It is known that harmonics functions can be combined into a sine using the following:

$$\sum_{i=1} A_i \sin(i\phi) + \sum_{i=0} B_i \cos(i\phi) = \sum_{i=1} A'_i \sin(i\phi + \phi i0) + Constant. \tag{H.1}$$

Here we find constant is contributing less than 2% of the combined function in any case, so we drop it. We find the amplitude of first and second order of combined sin function has a ratio about 4:1 when field strength is near zero. This ratio is used in the minimal model presented in the main text.

Another interesting result is when we study the ground state phase $\phi_{i0}$ in the first and second harmonics, we find they are increasing linearly with $B_x$ field within the range 0-100mT. This was mentioned in [6] but here we provide more details. Based on the simulation, we get $\phi_{20} > 2\phi_{10}$ and the grey shadow region is indicating the $\delta_{12}$ increase in with field. Hence we can confirm there is a $\delta\phi$ term in the CPR from the simulation. How this $\delta\phi$ related to the strength of spin-orbit interaction is worth a further discussion in future works.

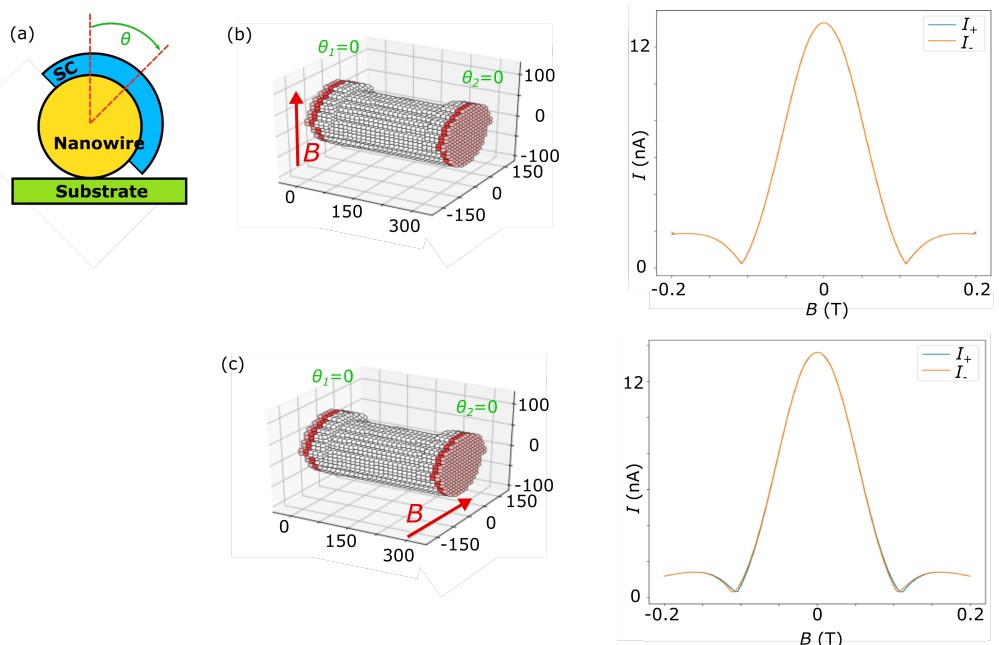

Figure 22: (a)Diagram of cross-section of InSb nanowires (Nanowires) that are half-covered by the superconductor Tin (SC). $\theta$ is indicating the angle between center of the shell and the normal direction of the substrate. (b) Left: diagram of how field is applied when both shell is placed at $\theta = 0$ degree. Right: Supercurrent diffraction pattern when field is applied along out of plane direction. (c) Right: Supercurrent diffraction pattern when field is applied along in plane but perpendicular direction.

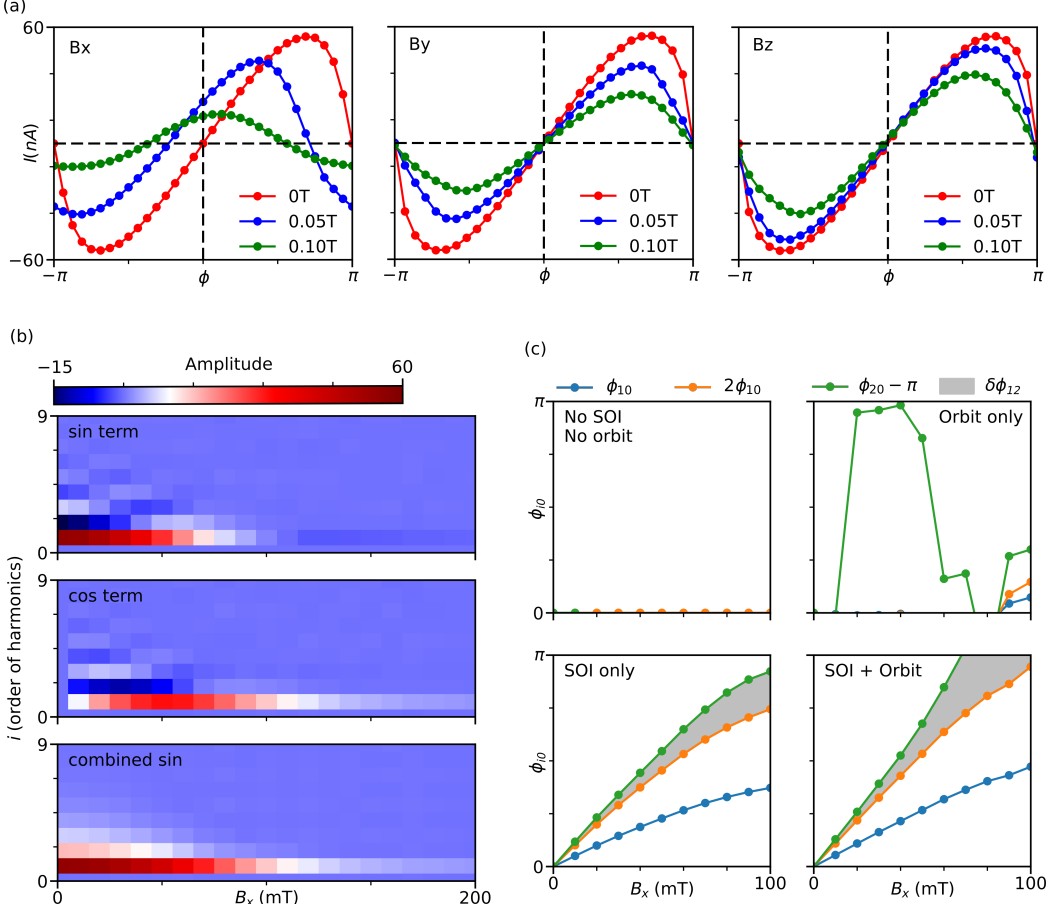

Figure 23: (a) CPR when external field is along three directions related to the device at field strength equals to 0T, 0.05T and 0.1T. (b) Amplitude of Sine and Cosine terms that are derived from Fourier expansion of CPR when external field is along $\hat{x}$-direction. The combined sine term is plotted as function of $B_x$ field and order of harmonics. (c) Ground state phase $\phi_{n0}$ at the first and the second order of combined sin harmonics as function of external field $B_x$. A constant $\pi$ was subtracted from all second order harmonics but has no effect as second order harmonic has a period of $\pi$.

# I   Temperature dependence of skewed diffraction patterns

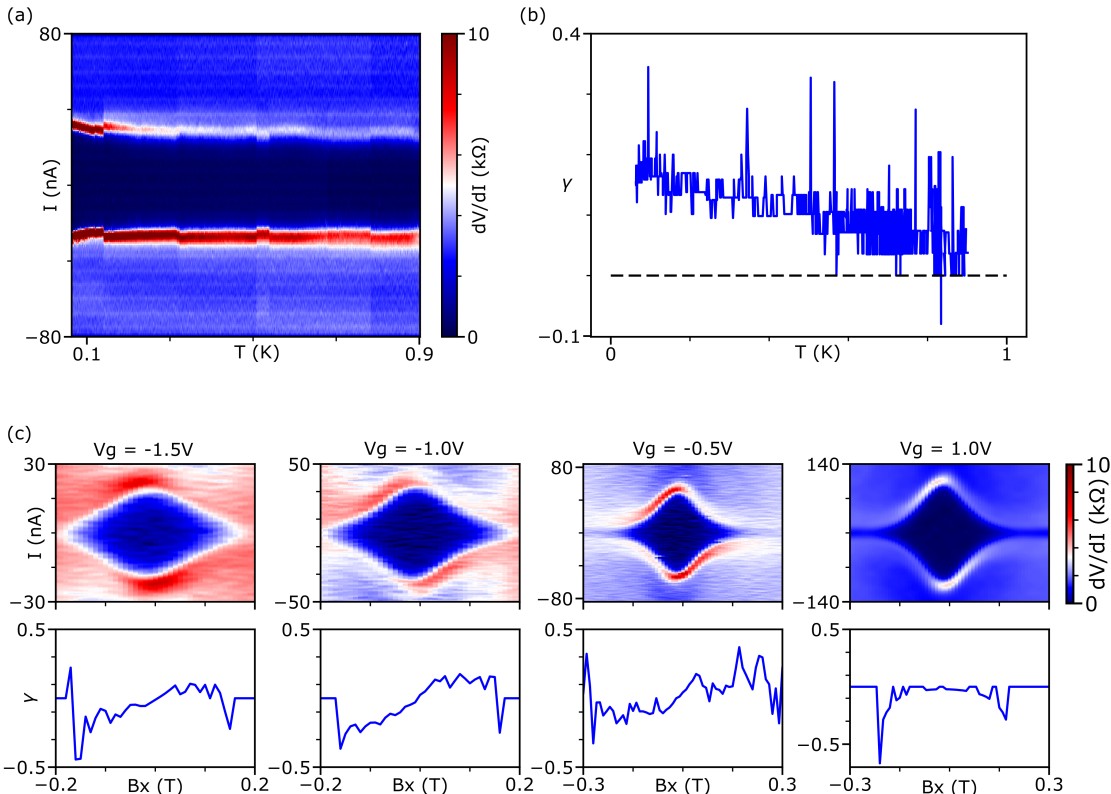

Figure 24:  Temperature dependence in Device A. (a) For $V_{gate} = -2V$ and external field $B_x = 50mT$. dV/dI differential resistance is plotted as function of current source I and temperature T. (b) Coefficient $\gamma$ is plotted as function of temperature T. (c) Diffraction patterns taken at T=1.1K, $\gamma$ is extracted from 2D scan results and plotted as function of $B_x$.

# J  $\gamma$ trace in different field

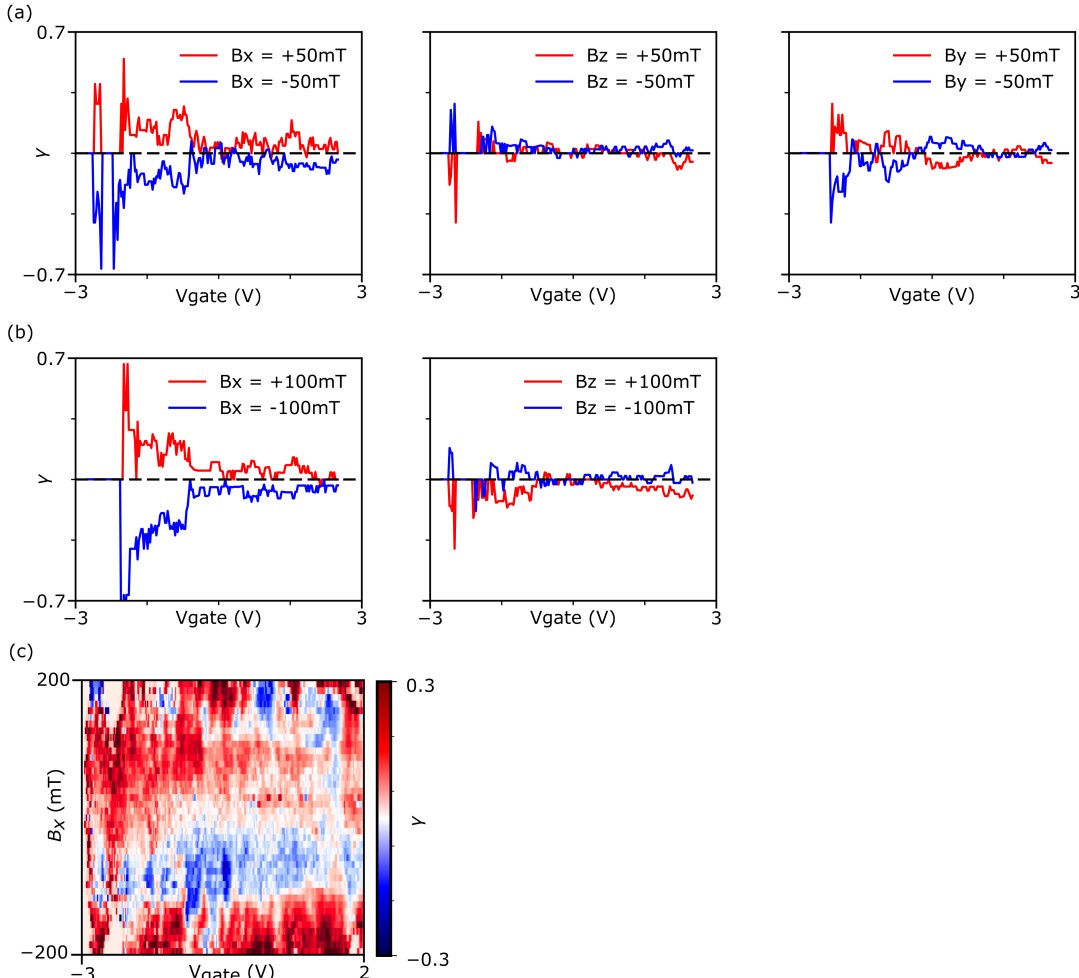

Figure 25: Coefficient $\gamma$ as function of $V_{gate}$ in different external field $B$. (a) External field with fixed strength $|B| = 50mT$ and along $\hat{x}$, $\hat{y}$, and $\hat{z}$ axis. (b) External field with fixed strength $|B| = 100mT$ and along $\hat{x}$ and $\hat{z}$ axis. (c) The 2D $B_x$ versus $V_{gate}$ map of coefficient gamma of Device C. The gate voltage range is corresponding to the chemical potential $\mu = 6 - 30meV$.

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
