# Peer review of "Evidence of $φ$0-Josephson junction from skewed diffraction patterns in Sn-InSb nanowires"

_SciPost Physics, doi:SciPost Phys. 18, 013 (2025)_

## Round 2 · Referee Report · Anonymous (Referee 1) · 2023-9-14

Report

The authors analyze the critical current behavior for a hybrid Sn/InSb junction in a magnetic field. They point out an asymmetry in the forward and reverse current which they associate with phi0 behavior of the junction current phase relation. Their claim is supported by some model calculations including spin-orbit interactions.

In my opinion, the results presented in this manuscript are sound. However, I have some concerns with respect to the authors interpretation and the way that the results are presented.

First of all, while the authors focus on phi0 behavior as the cause of the asymmetric critical current patterns in a magnetic field, it should be stressed that an anomalous phase shift is by itself not enough to produce such asymmetries. Rather than that, the asymmetric patterns require a diode effect, which is not necessarily linked to the phi0 behavior. While this is implicit in the discussion around Eq. (2), I find that the title and the abstract are misleading as a pure phi0 effect can only be detected through phase biased measurements.

On the other hand, phi0 behavior in these type of junctions in a magnetic field is not surprising and have been reported in previous works like Ref. 34 in the manuscript (see also https://arxiv.org/abs/2208.11198 for more recent experiments based on microwave spectroscopy in phase biased InAs junctions). In contrast, the superconducting diode effect have been observed mainly in van der Waals junctions and, to my knowledge, it is at present not fully understood.

For these reasons I believe that the manuscript requires a thorough revision in order to clarify the difference with previous works in which phi0 behavior has been reported.

In addition, I have a number of comments that should be addressed before publication:

  • In device A the junction is placed closer to the left Au lead. For this reason the Sn shell on the left could be more affected by inverse proximity effect than the right one. Although it is not likely that this could have an influence on the supercurrent asymmetry it could be relevant for the junction transport properties and should be commented.

  • The supercurrent is measured in a two terminal configuration. It would be worth that the authors comment on the method used to eliminate the possible effect of any series resistance and its size.

  • Regarding the tight-binding model calculations, the size of the spin orbit parameter which is necessary to get an effect of the same order as in experiments is extremely large compared to existing estimates. I wonder whether this could be pointing out to some missing ingredient in the modeling. It would be convenient that the authors give more details on these calculations.

  • On the other hand I have some concerns regarding the paper organization. For instance, the device description in Figure 2 is then repeated in the description of Figure 4. I also don't understand the importance given to the "phenomenological model" which is just a mathematical expression of a current phase relation with two displaced harmonics.

---

## Round 2 · Referee Report · Antonio Manesco (Referee 2) · 2023-9-20

Strengths

1- The authors show that the phenomenon reported does not result from fine-tuning. 2- Code and data are fully available, with datasets containing data beyond the manuscript's content. 3- The authors provide a theoretical model that mostly reproduces the measured data qualitatively. 4- The manuscript has a clear message and the story is coherent.

Weaknesses

1- The microscopic model can only reproduce quantitatively the observed phenomenon if the spin-orbit strength is overestimated. 2- Some discrepancies between the tight-binding simulations and experimental results are not discussed.

Report

The authors report Josephson current measurements in Sn-InSb nanowires. The results show evidence of $\phi_0$-Josephson effect. In the manuscript, the authors provide a phenomenological explanation backed up with microscopic simulations to interpret their measurements. Furthermore, the authors performed parameter sweeps to ensure that the observed effect requires no fine-tuning.

I list below a series of points that I consider relevant to be addressed before publication.

  1. The so-called "phenomenological model" is not more than a harmonic decomposition of the critical current. I understand that it gives an intuitive explanation to the reader, and is also backed up with previous works suggesting that the decomposition is sound. But I do not think it is appropriate to call it a "model". If the authors are willing to name it a phenomenological model, I would expect a relation between spin-orbit field direction and the parameters in Eq. (1). Otherwise, naming it "harmonic decomposition" or something similar sounds more appropriate and requires no further changes in the manuscript.
  2. I suggest, if the authors consider it appropriate, to bring Fig. 2 to the beginning of the manuscript to present the device layout early on.
  3. The authors mention that the tight-binding model shows $\gamma=0$ in the absence of Zeeman field. However, figure 2a shows $\gamma\neq0$ for $B_x=0$. This result does not match the tight-binding results shown in the same figure (panel 2b). Can the authors provide an explanation for this discrepancy?
  4. The authors say: "Orbital effect only ($\alpha = 0,~A\neq 0$) yields a similar structure (see $B_x = 80\mathrm{mT}$), limited in magnitude and field. We believe this is a simulation artifact that appears when the external field is perpendicular to the line connecting the center of the wire and the center of the shell. However, there is no explanation to justify that the results are a simulation artifact." Can the authors expand on this statement and/or provide numerical evidence that the statement is sound?
  5. Since the authors mention that the simulated device has the same geometry as the measured nanowires and all the parameters are chosen to match the experiments, why is it necessary to overestimate the spin-orbit coupling? Could that hint that SOC is not the main cause of the phenomenon?
  6. In both the experiments and the simulations it's possible to observe finite $\gamma$ even when $\mathbf{B} \perp \hat{x}$. I understand that it is likely hard to resolve the direction of the spin-orbit field in the experiments. But I am wondering why the simulations show a similar feature.

The following comments are about the Supplementary Materials.

  1. In Sec. V.B the authors report hysteresis as a function of bias voltage. Can the authors provide an explanation for the presence of this hysteresis?
  2. In Sec. VI, the authors say that the gauge is fixed so that the system "is [translationallly?] invariant along the $x$-direction". However the gauge choice explicitly depends on $x$, so I don't follow the statement. Do you rather mean that the field is perpendicular to $\hat{x}$?
  3. In Sec. VI, the authors say that "kwant.continuum.discretize cannot handle the systems with lower symmetry", however the tutorial shows an example with no translational symmetry. It is unclear to me what the authors mean by this statement.

Requested changes

1- Explain the discrepancies between the two panels in Fig. 2. 2- Avoid calling Eq. (1) a phenomenological model, or relate the parameters in the equation with system parameters. 3- Appropriately address the questions in the referees' reports.

---

## Round 3 · Referee Report · Anonymous (Referee 1) · 2024-3-9

Report

I agree with the authors that the term "diode" gives the wrong impression that current is fully suppressed in one direction. What I asked them is to clarify that a simple phi0 anomaly in the CPR is not enough to get the asymmetric skewness in the critical current vs field patterns that they observe. More than evidence of phi0 behavior what they find is evidence of an asymmetric CPR.

On the other hand the two referees agree on criticizing the "phenomenological model" and pointing out the extremely large value of the spin-orbit coupling needed in the tight-
binding calculations as indicating another possible source for the effect. None of these observations have led to revisions in the manuscript. In the discussion of the phenomenological model I would also point out that there must be some missprint in the expression $I_1,I_2 \propt (1- B/B^2_c)$.

Requested changes

Revise the manuscript in accordance to the referee reports

  • validity: -
  • significance: -
  • originality: -
  • clarity: -
  • formatting: -
  • grammar: -

Author:  Bomin Zhang  on 2024-06-08  [id 4549]

(in reply to Report 1 on 2024-03-09)

The authors thank all referees for their comments. We will be posting replies individually on SciPost. Because there is no color difference, we quote the referee’s comments with ‘>” and our response with ‘>>>’

Referee 1:

I still disagree that the "phenomenological model" is a model. The data visibly consists of a periodic pattern with higher-order harmonics and an offset from zero. Equation 1 could describe the data regardless of the physical meaning of the x- and y-coordinates. It does not have any connection with system parameters, which is what I would consider necessary to call it a model. However, I believe that the wording does not change the reported results and readability. Thus, I do not think that the author's decision on the subject should affect acceptance for publication.

I however am still curious about the large value of spin-orbit coupling used in the simulations. The authors are right that the exact value used in the simulations is not supposed to be quantitatively accurate -- to some extent. There should be still a relevant energy/length scale that relates the SOC to the reported phenomenon. The understanding of the relevant scales would not only help to justify the overestimated values of SOC, but also obtain further intuition of the phenomenon. In fact, this is what I would expect to understand from a phenomenological model. Can the authors estimate the magnitude of γ based on relevant energy/length scales?

We appreciate the referees' inspired suggestions. And we have plans to continue to investigate this effect so we can better relate the strength of SOC to the reported phenomenon. However, we don’t have enough data from experiments and simulations to make the estimation.

To relate an offset between two CPR harmonics to the SOI strength, which is pretty well known by now in these wires, is not a very direct process and cannot be done at the back of the envelope level. Anything else involves numerous system parameters that are not precisely known. For comparison, in similar wires some of us have related measurements of quantum dot states to SOI parameters and that is possible from a very simple heuristic that relates SOI length to dot size. Those are probably some of the most direct estimates, because you see the anticrossing of spin states themselves, and it is due to SOI. Other methods, such as WAL, are not very reliable at least when it comes to nanowires. This delta-offset is of the same kind.

Related to the previous point: the effects of the orbital field are now conjectured to be related to chiral-induced magnetism. I thus have two questions:

  1. Does the theoretical framework for the tight-binding simulation include the magnetism effects required for the conjectured origin of the orbital effect?

The relevant chirality can come from junction geometry, as long as that is not inversion symmetric. We are investigating this for a future study.

  1. The orbital effects seem comparable to SOC in Fig. 5b. Are the orbital contributions the dominant one if SOC is not overestimated? Would that hint at an alternative origin of the observed phenomenon?

The “orbital” effect is quite limited, related to a small feature in the Ic vs B characteristic. It also does not possess the field anisotropy of the SOC effect. So, while it is curious that this comes up in the simulation and does deserve a separate study, we do not feel it is a significant enough effect to change our preferred interpretation.

Referee 2:

I agree with the authors that the term "diode" gives the wrong impression that current is fully suppressed in one direction. What I asked them is to clarify that a simple phi0 anomaly in the CPR is not enough to get the asymmetric skewness in the critical current vs field patterns that they observe. More than evidence of phi0 behavior what they find is evidence of an asymmetric CPR.

This is a matter of framing the results, and we prefer our framing. Some of us have worked for many years on trying to unambiguously establish the phi0 behavior. The “asymmetric CPR”, in our view, provides a way to do this that is more robust to alternative explanations of phase shifts due to induced flux. Some of us find the phi0 effect more interesting than SDE.

On the other hand the two referees agree on criticizing the "phenomenological model" and pointing out the extremely large value of the spin-orbit coupling needed in the tight- binding calculations as indicating another possible source for the effect. None of these observations have led to revisions in the manuscript.

Referee 1 is now okay with us proceeding with our phenomenological model.

As for the tight-binding model, we do not think that this model is quantitatively accurate, even if it does make an effort to represent some of the key features of the samples. We disagree that the value we used is “extremely” large, it is nominally larger than what we know to be the SOI strength in InSb nanowires, but it is just a model parameter. We do not think it is necessary or easy to develop a quantitatively accurate model.

Our main argument why SOI is likely the main source of skew is the experimental observation of the magnetic field orientation dependence of the skew coefficient, which is captured by the tight-binding model.

In the discussion of the phenomenological model I would also point out that there must be some missprint in the expression I1,I2 \propt(1−B/Bc^2)

We thank the referee for finding this typo. This is corrected now. .

---

## Round 3 · Referee Report · Antonio Manesco (Referee 2) · 2024-3-11

Report

I still disagree that the "phenomenological model" is a model. The data visibly consists of a periodic pattern with higher-order harmonics and an offset from zero. Equation 1 could describe the data regardless of the physical meaning of the x- and y-coordinates. It does not have any connection with system parameters, which is what I would consider necessary to call it a model. However, I believe that the wording does not change the reported results and readability. Thus, I do not think that the author's decision on the subject should affect acceptance for publication.

I however am still curious about the large value of spin-orbit coupling used in the simulations. The authors are right that the exact value used in the simulations is not supposed to be quantitatively accurate -- to some extent. There should be still a relevant energy/length scale that relates the SOC to the reported phenomenon. The understanding of the relevant scales would not only help to justify the overestimated values of SOC, but also obtain further intuition of the phenomenon. In fact, this is what I would expect to understand from a phenomenological model. Can the authors estimate the magnitude of γ based on relevant energy/length scales?

Related to the previous point: the effects of the orbital field are now conjectured to be related to chiral-induced magnetism. I thus have two questions: 1. Does the theoretical framework for the tight-binding simulation include the magnetism effects required for the conjectured origin of the orbital effect? 2. The orbital effects seem comparable to SOC in Fig. 5b. Are the orbital contributions the dominant one if SOC is not overestimated? Would that hint at an alternative origin of the observed phenomenon?

Requested changes

Relate system parameters with the magnitude of γ. That will (if the authors' interpretation is correct) both justify that the overestimated value of SOC is reasonable and rule out the importance of orbital effects.

---

## Round 3 · Author Response

The authors thank all referees for their comments. We will be posting replies individually on SciPost. Because there is no color difference, we quote the referee’s comments with ‘>” and our response with ‘>>>’.

Report 1:

  1. The so-called "phenomenological model" is not more than a harmonic decomposition of the critical current. I understand that it gives an intuitive explanation to the reader, and is also backed up with previous works suggesting that the decomposition is sound. But I do not think it is appropriate to call it a "model". If the authors are willing to name it a phenomenological model, I would expect a relation between spin-orbit field direction and the parameters in Eq. (1). Otherwise, naming it "harmonic decomposition" or something similar sounds more appropriate and requires no further changes in the manuscript.

It is a model because it describes the data, namely Isw vs B, from a postulated junction current-phase relation and postulated parameter dependences in it, e.g. phase shifts on B. We are not decomposing Isw vs B experimental data so the word ‘decomposition’ is not applicable.

  1. I suggest, if the authors consider it appropriate, to bring Fig. 2 to the beginning of the manuscript to present the device layout early on.

Fig 2 is in the beginning of the manuscript, it is in the front 1/3 of the main text figures and before the supplemental sections.

  1. The authors mention that the tight-binding model shows gamma = 0 in the absence of Zeeman field. However, figure 2a shows gamma !=0 for B_x = 0. This result does not match the tight-binding results shown in the same figure (panel 2b). Can the authors provide an explanation for this discrepancy?

We believe the referee is referencing Figure 6 instead of Figure 2. Fig 6a shows a blue stripe for positive fields and a red stripe for negative fields. In between, near zero, the values are low. It is true that they are not zero, which is an artifact of how the field was swept. In our most careful scans gamma=0 at B=0. We explain the nature of the artifacts below,and mention it now in the main text at Section: Figure 6.

Our methodology for measuring the coefficient gamma involved sweeping the gate voltage dependence at each magnetic field setting, this process generated approximately 200 data sets. From these, we extracted the gate voltage dependence of gamma and compiled the results for comparative analysis (Figure. 6B). Despite making the scans slow to ensure accuracy, we found the field hysteresis is unavoidable, which is attributable to the residual magnetic field in the magnet.

Field hysteresis is a common phenomenon in transport measurement from the instruments. To address this, we have provided a discussion on the effects of field hysteresis in Figure. S10. In this figure, we explain how this hysteresis influences our measurements and its impact on our results.

  1. The authors say: "Orbital effect only (alpha = 0, A != 0) yields a similar structure (see B_x = 80mT), limited in magnitude and field. We believe this is a simulation artifact that appears when the external field is perpendicular to the line connecting the center of the wire and the center of the shell. However, there is no explanation to justify that the results are a simulation artifact." Can the authors expand on this statement and/or provide numerical evidence that the statement is sound?

We added sentences and references in section: Tight-Binding model and a figure in supplementary to explain this effect. When there is no inversion symmetry about the external field direction, we can observe this skewed pattern. However, we believe this effect worth further discussion with a full new project, so we didn’t put full simulation results there.

  1. Since the authors mention that the simulated device has the same geometry as the measured nanowires and all the parameters are chosen to match the experiments, why is it necessary to overestimate the spin-orbit coupling? Could that hint that SOC is not the main cause of the Phenomenon?

Regarding the value of spin-orbit coupling, we do not think that this tight-binding model is quantitatively accurate, even if it does make an effort to represent some of the key features of the samples. So in itself, the fact that there is a quantitative discrepancy in coefficients chosen does not indicate the presence of other factors.

At the same time, we are open to learning about other possibilities and will continue investigating them.

We have included a statement in the main text:” Some of the parameters, such as spin-orbit strength, exceed those previously reported for InSb nanowires. However, we cannot claim that matching this model to data is a reliable way of extracting spin-orbit interaction strength.”

  1. In both the experiments and the simulations it's possible to observe finite gamma even when B⊥x. I understand that it is likely hard to resolve the direction of the spin-orbit field in the experiments. But I am wondering why the simulations show a similar feature.

We added a new reference in the main text (Ref. 44) for the orbital related unequal supercurrent in each source bias. This work also discussed the chirality induced magnetism when field is parallel to the current direction, which can explain the finite gamma we observed in the experiment and simulation.

  1. In Sec. V.B the authors report hysteresis as a function of bias voltage. Can the authors provide an explanation for the presence of this hysteresis?

Hysteresis is most commonly observed in underdamped Josephson junctions.

  1. In Sec. VI, the authors say that the gauge is fixed so that the system "is [translationallly?] invariant along the x-direction". However the gauge choice explicitly depends on x, so I don't follow the statement. Do you rather mean that the field is perpendicular to x-direction?

In this paper, we consider a uniform magnetic field along an arbitrary direction B = [B_x,B_y,B_z]. Due to the redundant gauge degrees of freedom, we can choose the vector potential A = [0, B_zx,B_xy-B_y*x] so that it is invariant along the nanowire (i.e., z-direction). There is a typo in the manuscript due to different coordinates are used in our work compare to that in ref.43, and we apologize for the confusion.

  1. In Sec. VI, the authors say that "kwant.continuum.discretize cannot handle the systems with lower symmetry", however the tutorial shows an example with no translational symmetry. It is unclear to me what the authors mean by this statement.

Our model possesses translational symmetry when there is no spin-orbital coupling. So, we can use the discretize method to construct the nanowire. However, the technical details of this tutorial (https://kwant-project.org/doc/1/tutorial/discretize) also state that "The builder returned by discretize will have an N-D translational symmetry, where N is the number of dimensions that were discretized. This is the case, even if there are expressions in the input (e.g. V(x, y)) which in principle may not have this symmetry. When using the returned builder directly, or when using it as a template to construct systems with different/lower symmetry, it is important to ensure that any functional parameters passed to the system respect the symmetry of the system. Kwant provides no consistency check for this."
Thus, we need to manually add the Peierls substitution to the hopping term when considering the SO effects because it breaks the translational symmetry. Here, we just want to emphasize that we shouldn't discretize the Hamiltonian that contains the SO effect directly.

Referee 2:

First of all, while the authors focus on phi0 behavior as the cause of the asymmetric critical current patterns in a magnetic field, it should be stressed that an anomalous phase shift is by itself not enough to produce such asymmetries. Rather than that, the asymmetric patterns require a diode effect, which is not necessarily linked to the phi0 behavior. While this is implicit in the discussion around Eq. (2), I find that the title and the abstract are misleading as a pure phi0 effect can only be detected through phase biased measurements.

phi0 effect cannot be reliably detected through phase-biased measurements because it requires the application of very large magnetic fields, and these fields will introduce a phase bias into the loop. This bias cannot be reliably distinguished from the phi0 shift in the current-phase relation. Even gate-tunable shifts in SQUID patterns are susceptible to this problem, because by changing the gate in the presence of a large field, the effective size of the loop and the flux threading it are changed.

The asymmetry studied here therefore provides a remarkably robust way of identifying the presence of the intrinsic phi0 shift. The term ‘diode’ is misguided because it implies some sort of useful applications and also a dramatic asymmetry, namely no supercurrent in one bias direction. This is not present in these superconducting devices. So we are not interested in using this term.

On the other hand, phi0 behavior in these type of junctions in a magnetic field is not surprising and have been reported in previous works like Ref. 34 in the manuscript (see also https://arxiv.org/abs/2208.11198 for more recent experiments based on microwave spectroscopy in phase biased InAs junctions). In contrast, the superconducting diode effect have been observed mainly in van der Waals junctions and, to my knowledge, it is at present not fully understood.

It was reported but it was not proven. We believe our work proves it. See above for the discussion of the ‘diode’.

For these reasons I believe that the manuscript requires a thorough revision in order to clarify the difference with previous works in which phi0 behavior has been reported.

The manuscript is framed and phrased in a way that reflects our understanding of the problem. In particular, it contains a thorough discussion of the points made by the referee already. Our interest in this topic and the search for how to demonstrate the phi0-junction effect predates the emergence of the new ‘diode’ terminology and will probably outlive it.

In device A the junction is placed closer to the left Au lead. For this reason the Sn shell on the left could be more affected by inverse proximity effect than the right one. Although it is not likely that this could have an influence on the supercurrent asymmetry it could be relevant for the junction transport properties and should be commented.

Indeed junctions made of two different superconductors still show symmetric current-voltage characteristics. Nanowire junctions, due to their small dimensions, are always asymmetric in their geometry.

Critical currents in device A reach 200 nA for positive gate settings, which is typical for Sn-InSb-Sn junctions (see ref 39), so we do not find evidence that superconductivity is suppressed by inverse proximity effect from the leads

The supercurrent is measured in a two terminal configuration. It would be worth that the authors comment on the method used to eliminate the possible effect of any series resistance and its size.

We acknowledge the referee’s request for clarification regarding how series resistance is accounted for in our data. In response, we propose to add the following explanation in Supplementary Section I:

“ Transport 2-point measurements with currents source and voltage measurement model in parallel are used, with several stages of filtering placed at different temperatures. (add sentence) The series resistance in our measurements arises from several sources: the filters in the measurement circuit, the voltage measurement model, and the contact resistance. However, given that the Sn shell is in situ grown on the nanowires, forming transparent contacts as detailed in Ref. 39, we primarily consider the resistance contributions from the measurement setup. Specifically, the resistance from the RC filters in 2-point measurement amounts to 4.04 Kohm, while the voltage measurement model contributes an additional 3 Kohm. These values are considered in our analysis to ensure precise and reliable data interpretation, and a series resistance of 7.04 kohm is subtracted from all 2point measurement devices.”

Regarding the tight-binding model calculations, the size of the spin-orbit parameter which is necessary to get an effect of the same order as in experiments is extremely large compared to existing estimates. I wonder whether this could be pointing out to some missing ingredient in the modeling. It would be convenient that the authors give more details on these calculations.

‘Extremely large’ is a relative term. We do not think it is ‘extremely large’. For instance in this work spin-orbit is overextracted by a factor of 100: https://www.nature.com/articles/s41467-017-00315-y . The paper and its analysis of SOI were subsequently refuted here: https://pubs.acs.org/doi/full/10.1021/acs.nanolett.7b03854 .

Regarding the value of spin-orbit coupling, we do not think that this tight-binding model is quantitatively accurate, even if it does make an effort to represent some of the key features of the samples. So in itself the fact that there is a quantitative discrepancy in coefficients chosen does not indicate the presence of other factors.

At the same time, we are open to learning about other possibilities and will continue investigating them.

We have included a statement in the main text:” Some of the parameters, such as spin-orbit strength, exceed those previously reported for InSb nanowires. However, we cannot claim that matching this model to data is a reliable way of extracting spin-orbit interaction strength.”

Detailed information about our numerical model can be found in Section VI of the Supplementary material, and all relevant codes are available in Zenodo (Ref. 44).

On the other hand I have some concerns regarding the paper organization. For instance, the device description in Figure 2 is then repeated in the description of Figure 4.

We value the referee’s feedback regarding the repeated device schematic. However, we believe its presence is crucial for aiding readers in comprehending the orientation of the external field relative to the device. Therefore, we prefer to keep these schematics in the figures to ensure that the context and setup of measurements are clearly communicated to our audience.

I also don't understand the importance given to the "phenomenological model" which is just a mathematical expression of a current phase relation with two displaced harmonics.

It is not just that, the model also includes the dependence of parameters of the current-phase relation on the field. It reproduces Isw vs B data.

The importance is that the model works, using simple assumptions.

---

## Round 3 · List of Changes

1. Add sentences and reference 44 in Section: Tight-binding model in the main text to explain skewed pattern (unequal supercurrents) due to orbital effect only.

  2. Add Section: VII. D in supplementary, together with FigS16 to explain how orbital effect in simulation is introduced due to break of inversion symmetry.

  3. Added Sentences in Supplementary Section I to explain how series resistance is substracted in 2-points measurements.

  4. Coordinates used in the Section VI is corrected.

---

## Round 4 · Referee Report · Antonio Manesco (Referee 2) · 2024-6-21

Report

The referee's addressed most of my criticism, except for choosing their "preferred interpretation" for the origin of the $\phi_0$ effect. Particularly, I would like to refer to [Davydova et al., Sci. Adv. 8, eabo0309 (2022)], where the authors propose a purely orbital origin for the anomalous Josephson effect. I expect this contribution to also depend on the direction of the magnetic field. Moreover, a recent work [Reinhardt, S., Ascherl, T., Costa, A. et al. Nat Commun 15, 4413 (2024)] argues that whereas an anomalous Josephson effect caused by SOC has a strong gate dependence, the orbital counterpart is mostly insensitive to the gate voltage. Since the authors claim that "effect is observed over wide ranges of gate voltage" and that "no consistent effect of gate voltage on the skew magnitude is observed", their observations seem consistent with orbital effects. Thus, it remains unclear to me why the authors "do not feel it [orbital effect] is a significant enough effect to change our preferred interpretation".

I am keen to recommend the manuscript for publication if the authors provide enough data to justify their preferred explanation or give appropriate weight for alternative explanations.

Recommendation

Ask for minor revision

---

## Round 4 · Referee Report · Anonymous (Referee 1) · 2024-7-2

Report

The authors have responded to my previous comments, but they have not addressed the clarifications I requested regarding the difference between Phi0 behavior and asymmetric CPR, nor have they addressed my criticism on the phenomenological model. Arguments based on authority, such as "Some of us have worked for many years on trying to unambiguously establish the Phi0 behavior" or personal preference, such as "Some of us find the Phi0 effect more interesting than SDE" do not adequately address my comments. I find that the authors are not being sufficiently receptive to my constructive feedback. Thus, after two refereeing rounds, I still cannot recommend publication of the manuscript in the present form but leave the final decision to the editors.

Recommendation

Ask for major revision

---

## Round 4 · List of Changes

Delete sentence from the main text: "Using realistic junction parameters, the numerical model is capable of reproducing the key experimental observations. ", which is a too strong claim.
Modified eqaution in the main text , from $I_{1}, I_{2} \propto (1-{B}/{B_c}^{2})$ to $I_{1}, I_{2} \propto (1-{B}^{2}/{B_c}^{2})$. The original equation has a typo.

---

## Round 5 · Referee Report · Anonymous (Referee 3) · 2024-10-26

Report

I read the remaining criticism of both referees:

Referee 1 sees a stronger evidence for an orbital effect as opposed to a spin-orbit coupling as origin of the effect. The author‘s statement, that there is no SYSTEMATIC gate dependence, is taken as an absence of a gate dependence. Figure 6, however, displays a clear, but fluctuating gate variation of the diode efficiency, which is qualitatively reproduced by model calculations. I do not necessarily expect the same systematic gate dependence as in the cited work by Reinhardt et al., because the potential landscape in the nanowires is typically much more disordered when compared to 2DEGs. On the other hand, there seem to be no reasons for such fluctuations in the orbital effect proposed by Davydova et al. Hence, I support the author‘s point of view.

Referee 2 requests a clear discrimination between phi0-shift on one hand and anharmonic CPR on the other. In their discussion of Eq. 1, the authors clearly state that the diode effect results from the difference between the phi0-shifts of the first two harmonics of the CPR, while a sinusoidal CPR produces no diode effect. Even though their experiment is not sensitive to the global phi0-shift, it detects the relative shift \delta_12 via the diode effect. In my view, this is correct and clear enough.

For these reasons, I recommend acceptance of the manuscript in its present form.

Recommendation

Publish (easily meets expectations and criteria for this Journal; among top 50%)

---

## Round 5 · Author Response

The authors thank all referees for their comments. We will be posting replies individually on SciPost. Because there is no color difference, we quote the referee’s comments with ‘>” . Report 1:

The referee’ss addressed most of my criticism, except for choosing their "preferred interpretation" for the origin of the Phi0 effect. Particularly, I would like to refer to [Davydova et al., Sci. Adv. 8, eabo0309 (2022)], where the authors propose a purely orbital origin for the anomalous Josephson effect. I expect this contribution to also depend on the direction of the magnetic field. Moreover, a recent work [Reinhardt, S., Ascherl, T., Costa, A. et al. Nat Commun 15, 4413 (2024)] argues that whereas an anomalous Josephson effect caused by SOC has a strong gate dependence, the orbital counterpart is mostly insensitive to the gate voltage. Since the authors claim that "effect is observed over wide ranges of gate voltage" and that "no consistent effect of gate voltage on the skew magnitude is observed", their observations seem consistent with orbital effects. Thus, it remains unclear to me why the authors "do not feel it [orbital effect] is a significant enough effect to change our preferred interpretation". I am keen to recommend the manuscript for publication if the authors provide enough data to justify their preferred explanation or give appropriate weight for alternative explanations. In fact, gate tunability of SOI in nanowires is not something we expect based on numerous previous measurements, the number for SOI comes out roughly the same. Any changes will be within the variations of our extracted skew. At the same time, spin-orbit anisotropy has been observed many times and the orientation of the anisotropy matches. The orbital effect would be comparable for either direction perpendicular to the nanowire while the spin-orbit effect is dramatically oriented perpendicular to the nanowire in the substrate plane. On a larger point, our paper provides both interpretations and gives arguments why we prefer the SOI one. It does not however suppress the other interpretation, and any reader can evaluate the results and their presentation and arrive at their own conclusions.

Report 2:

The authors have responded to my previous comments, but they have not addressed the clarifications I requested regarding the difference between Phi0 behavior and asymmetric CPR, nor have they addressed my criticism on the phenomenological model. Arguments based on authority, such as "Some of us have worked for many years on trying to unambiguously establish the Phi0 behavior" or personal preference, such as "Some of us find the Phi0 effect more interesting than SDE" do not adequately address my comments. I find that the authors are not being sufficiently receptive to my constructive feedback. Thus, after two refereeing rounds, I still cannot recommend publication of the manuscript in the present form but leave the final decision to the editors. We believe that the decision to refer to this phenomenon as a Phi0 junction or a superconducting diode should be left to the authors as a personal choice. We appreciate the discussion with the referee but wish to maintain the way we frame it in the paper.

---

## Round 5 · List of Changes

Add three references:
Ref. 19: J. S. Meyer and M. Houzet, Appl. Phys. Lett 125, 022603 (2024).
Ref.20 : S. Reinhardt, T. Ascherl, A. Costa, J. Berger, S. Gronin, G. C. Gardner, T. Lindemann, M. J. Manfra, J. Fabian, D. Kochan, et al., Nature Communications 15, 4413 (2024).
Ref.47: C. Owen and D. Scalapino, Physical Review 164, 538 (1967).

---

## Editorial Decision

published